# Marine oxygen production and open water supported an active nitrogen cycle during the Marinoan Snowball Earth

Benjamin W. Johnson [1,2], Simon W. Poulton [3] & Colin Goldblatt[1]

The Neoproterozoic Earth was punctuated by two low-latitude Snowball Earth glaciations. Models permit oceans with either total ice cover or substantial areas of open water. Total ice cover would make an anoxic ocean likely, and would be a formidable barrier to biologic survival. However, there are no direct data constraining either the redox state of the ocean or marine biological productivity during the glacials. Here we present iron-speciation, redox-sensitive trace element, and nitrogen isotope data from a Neoproterozoic (Marinoan) glacial episode. Iron-speciation indicates deeper waters were anoxic and Fe-rich, while trace element concentrations indicate surface waters were in contact with an oxygenated atmosphere. Furthermore, synglacial sedimentary nitrogen is isotopically heavier than the modern atmosphere, requiring a biologic cycle with nitrogen fixation, nitrification and denitrification. Our results indicate significant regions of open marine water and active biologic productivity throughout one of the harshest glaciations in Earth history.

---

[1] School of Earth and Ocean Sciences Bob Wright Centre A405, University of Victoria, PO Box 1700 STN CSC, Victoria, BC, Canada V8W 2Y2. [2] University of Colorado, Department of Geological Sciences UCB 399, Boulder, CO 80309-0399, USA. [3] School of Earth and Environment Maths/Earth and Environment Building, The University of Leeds, Leeds LS2 9JT, UK. Correspondence and requests for materials should be addressed to B.W.J. (email: bwjohnso@uvic.ca)

The late Neoproterozoic was a time of remarkable climatic and biological dynamism on Earth. After more than a billion years of stable climate through much of the Proterozoic, the Cryogenian period was punctuated by two long-lived, global Snowball Earth glaciations[1]. The appearance of multicellular organisms is thought to have occurred near this time period[2], as well as a rise in atmospheric oxygen in the Ediacaran[3]. Thus, some of the greatest climatic fluctuations and evolutionary innovations (e.g., multicellularity) occurred during this crucial interval.

It was initially postulated that a "hard Snowball Earth" with sea ice hundreds of metres thick covered the ocean during low-latitude glaciation[1,4]. This would have isolated the ocean from sunlight and atmospheric oxygen, presenting a formidable barrier to survival of oceanic species. However, no obvious mass extinction event is seen in the fossil record[5], though this is difficult to quantify given the lack of contemporary body fossils. Survivability solutions typically invoke "refugia" for organisms, ranging from global, low-latitude open-water belts[6] to sea-ice cracks[7] and ice shadows[8]. Geochemical calculations indicate small areas of open water would permit sufficient air sea-gas exchange for the atmosphere and surface ocean to equilibrate[9]. However, there are to-date no proxy data for the chemical state of the Snowball Earth ocean, and no direct evidence for biological productivity. In addition, the long duration (10's Myr) of the glaciations would allow for oxic weathering and reaction with metamorphic gases to deplete atmospheric oxygen completely if oxygenic photosynthesis was operating slowly or was absent.

To investigate the redox state of the Marinaon ocean, we utilised N-isotopes, Fe-speciation, and redox-sensitive trace elements (TE). The modern marine N cycles serves as a benchmark for interpretation of sedimentary rocks[10]. Currently, dissolved atmospheric $N_2$ is fixed to bioavailable N by single-celled organisms, with minimal isotopic fractionation of $\sim -2$ to $+2‰$[11,12] (Stable isotope notations are in per mil (‰) notation, where

$$\delta^X E(‰) = \left( \frac{^X E/^x E_{sample}}{^X E/^x E_{standard}} - 1 \right) \times 1000 \qquad (1)$$

E is element of interest, $X$ is heavy isotope, $x$ is light isotope. The standard used for $\delta^{15}N$ values is $N_2$ in air, which has a $\delta^{15}N$ value of 0‰ by definition.) fractionation can be larger with non Fe-Mo nitrogenase enzymes[13,14]. Nitrogen fixing organisms then die, release waste, or are consumed by other organisms, releasing N into the water column as $NH_4^+$. In oxygenated water, $NH_4^+$ is quickly converted to $NO_3^-$ via bacterial nitrification, leaving residual $NH_4^+$ isotopically heavier by $\sim 16‰$[15]. Complete nitrification does not preserve isotopic fractionation.

Nitrate can be converted to $N_2$ or $N_2O$ in the water column or sediments via denitrification ($NO_3 \rightarrow N_2$) or anammox ($NO_2 + NH_4 \rightarrow N_2$); both occur most rapidly in low-$O_2$ waters or pore waters[16], and leave residual $NO_3^-$ isotopically heavier by $>25‰$. Alternatively, dissimilatory nitrate reduction to ammonium can transform $NO_3^-$ to $NH_4^+$, again with no preserved isotopic effect if this goes to completion[17]. We use "denitrification" hereafter to include all pathways of $NO_3^-$ removal.

A small amount of biologically processed N sinks to the sediment. Without biologic cycling, little N would transfer to sediments and fixed N would likely have negative $\delta^{15}N$ values[18]. Organic N breaks down in sediment, releasing $NH_4^+$, which is absorbed by, or substitutes for $K^+$, in clay minerals. This process has little isotopic fractionation, especially in anoxic sediments[19]. Sedimentary N may faithfully record the isotopic signature of the water column from which it was deposited[10,20]. Since $\delta^{15}N$ values

depend on the balance of inputs (N fixation, minimal isotopic fractionation) and outputs (denitrification and anammox, large positive isotopic fractionation), sedimentary $\delta^{15}N$ indicates the state of the past N-cycle. In addition, as the balance of outputs depend on $O_2$ concentrations, sedimentary $\delta^{15}N$ is a proxy for the water redox state.

Though sedimentary $\delta^{15}N$ reflects water column N-cycling, different proportions of N-fixing to denitrification can result in equivocal sedimentary $\delta^{15}N$ values. For example, low, but positive, sedimentary $\delta^{15}N$ might record either quantitative (i.e., nearly 100%) or minimal water column denitrification. Quantitative denitrification would not preserve the 25‰ fractionation associated with conversion of $NO_3^-$ to $N_2$ or $N_2O$. No or small amounts of denitrification would result in water-column fixed N reflecting the isotopic value imparted by N-fixing, typically only 1–2‰ within atmospheric, assumed to be 0‰ during the Marinoan.

A further complication could arise in a fully anoxic ocean with ample $NH_4^+$. Ammonium assimilation has an isotopic fractionation, so biomass in $NH_4^+$-replete oceans should be isotopically negative[21]. Modern sediments generally reflect biologic cycling in the overlying water column[20], thus negative $\delta^{15}N$ values could record such an anoxic, but N-replete environment. If there is a deficit in dissolved $NH_4^+$, however, the fractionation from assimilation would not be preserved, as all available N would likely be assimilated into biomass. Complete assimilation of $NH_4^+$ in an anoxic ocean should result in biomass that is isotopically equal to fixed N. Again, we recognize the need for an independent redox proxy to assist in interpretation of N-isotopic data.

Since the amount of water column denitrification is directly related to the volume of anoxic water in the global ocean[16], independent redox proxies are required to break this degeneracy. Thus, we use iron-speciation[22] and redox-sensitive trace element[23] data to aid interpretation of N-isotopic data. Iron-speciation has proven to be a reliable indicator of local bottom water redox state, being sensitive to oxic, anoxic, and euxinic conditions. Uranium, V, and Mo are all more soluble in oxygenated water than in anoxic water; they are delivered via oxic weathering (i.e., require atmospheric oxygen), and their residence times are long ($>10^4$ yr) in oxic waters. In anoxic waters, U, V, and Mo may precipitate rapidly and are deposited to sediments. Under euxinic conditions, high Mo enrichments are often observed[23]. Thus, given a supply of TE to the ocean via oxic weathering, sediments which record no enrichment in these elements were likely formed beneath an oxic water column, those with moderate enrichments under low-$O_2$ to anoxic conditions, and those with especially high Mo enrichment under euxinic conditions.

We have recovered and analysed two sections of synglacial sedimentary rocks and one section of deglacial rocks representing the Marionan glacial (the second Neoproterozoic Snowball) from the Namibian Ghaub Formation. Using Fe-speciation, redox-sensitive trace element (TE), and nitrogen isotope geochemical data, we show that there was an active N-cycle, primary productivity, oxygen production, and an oxygenated atmosphere with at least pockets of oxygenated ocean water, despite extreme environmental stress.

## Results

### Geologic setting and sample description.
Neoproterozoic strata of Namibia[24] are characterised by marginal continental deposits, comprising a carbonate platform with occasional subaerial exposure, slope deposits, and deeper water units. Interspersed are two distinctive glacial units, the Sturtian Chuos and the

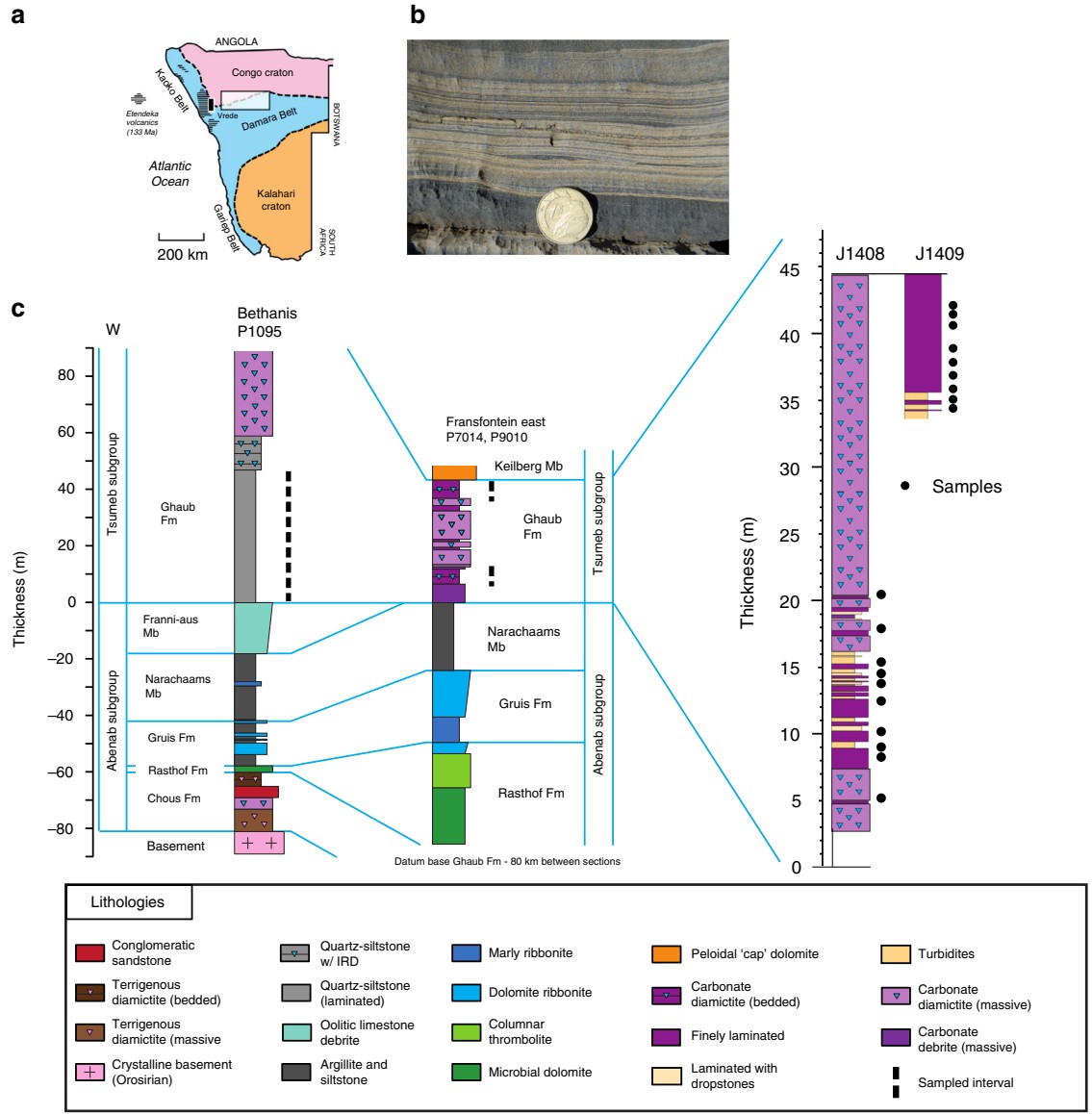

**Fig. 1** Geologic setting and sample locations. **a** Location map[49], **b** typical sample photo from section J1408, and **c** stratigraphy of studied sections. Fransfontein section is from ref. [24], and Bethanis section is from ref. [26]. Detailed section at Fransfontein highlights samples taken from only finely laminated beds without ice rafted debris or turbidites. Samples at Bethanis were taken at about 10 m spacing throughout the marked section. Sample section abbreviations are SGS (synglacial siliciclastic), SGC (synglacial with carbonate) and DGC (deglacial with carbonate)

Marinoan Ghaub Formations. Each glacial unit is overlain by a cap carbonate, the Rasthoff Formation and Keilberg Member, respectively. We have sampled and analysed samples from three sections of the Ghaub (Fig. 1). Importantly, all sampled intervals occur within the glacial interval, as evidenced by stratigraphic position below the uppermost diamictite (deposited synglacially) and cap carbonate, which indicates the end of Snowball glaciation[25].

The first sampled section is a synglacial siliciclastic (SGS) sequence from the Bethanis area[26]. The Ghaub Fm. in this area grades upward from a laminated quartz siltstone, with authigenic pyrite, into bedded laminations (sampled) with frequent ice-rafted debris (IRD), and finally into a massive carbonate diamictite. It is interpreted to represent an ice advance, though the timing of this advance during the glaciation is unknown. There is no evidence of wave action, so it is likely a deep water setting. This unit is synglacial, evidenced by minor IRD at the bottom of the section.

Other sampled sections are from Fransfontein ridge, 80 km to the east[24]. There are four facies at this location: massive carbonate diamictite, thinly laminated detrital carbonate with minor clay with and without IRD, and turbidites. Glacial activity in this region primarily eroded underlying carbonate platform units. The non-diamictite units at this section were likely deposited beneath a grounded ice sheet. Detrital carbonate was delivered, possibly by subglacial flow, into open water beneath floating ice. We sampled the thinly laminated beds without IRD, as this facies, synglacial with carbonate (SGC), should best represent ambient water conditions. The laminated facies are overlain by 30 m of diamictite.

Above the last glacial diamictite of SGC is the Bethanis member[24]. This unit represents the terminal deglaciation. It is a dark coloured, bedded, detrital Fe-rich dolostone with minor clay. It is conformably overlain by the Keilberg cap carbonate[24]. Again, we have sampled well-bedded intervals in an effort to analyze ambient water conditions during the deglacial (DGC).

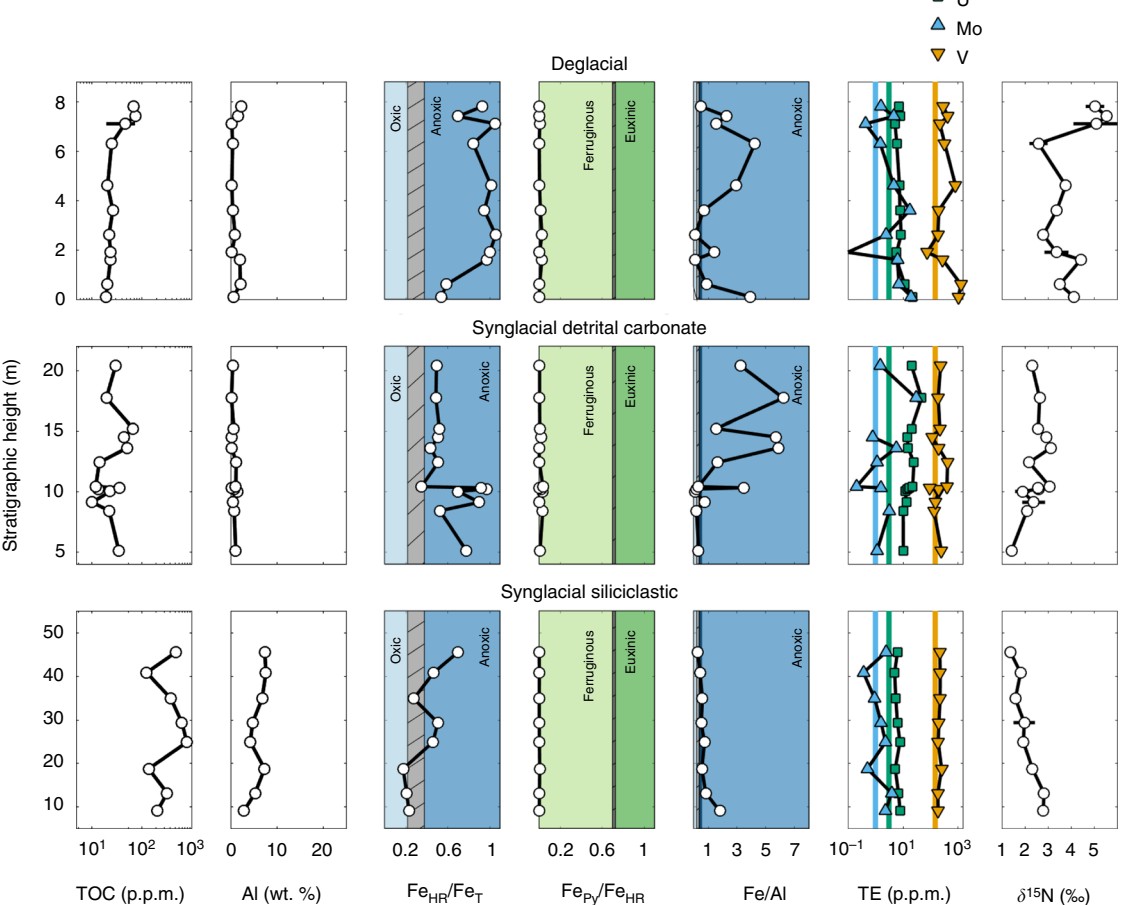

**Fig. 2** Geochemical results. Organic carbon (TOC), Al, Fe-speciation, Fe/Al, redox-sensitive trace element (TE), and $\delta^{15}N$ data shown for each section. TE are corrected for carbonate fraction (see text), all other data are whole rock. Iron speciation, with $Fe_{HR}/Fe_T<0.22$–$0.38$ and $Fe_{Py}/Fe_{HR}<0.7$, and Fe/Al ($>0.55\pm0.11$) indicate primarily anoxic, ferruginous conditions, which would be required for scavenging TE. TE accumulated in the water column as a result of oxic continental weathering. TE data are shown compared to Post Archean Average Shale (PAAS) values for each element (coloured lines). Nitrogen isotopes are also consistent with the presence of oxygenated surface water. Error bars are s.d. of three repeated analyses for $\delta^{15}N$

In all units, there is the potential that any measured biogeochemical data records reworked detrital material. Any such material could mask information about the water column during deposition. However, modern N in most marine settings mostly comes from sinking organic material sourced from primary production. Further, we find no correlation between N/Rb and $Al_2O_3$ (Supplementary Note 1), as would be expected if detrital influence was the main source of N to sampled sediments. In addition, previous work suggests the source of the majority of Ghuab sediments was the Ombaatjie Fm., representing a near-shore carbonate platform[24]. The Ombaatjie Fm. is relatively organic poor[27], and, thus, we suggest that detrital organic input is minimal in the sampled sections. Please see Supplementary Tables 1–4 and Supplementary Datasets 1–4 for full analytical results.

**Iron speciation indicates local bottom water anoxia.** Iron speciation data (Fig. 2, Supplementary Table 2) indicate similar bottom water conditions in all three sections. The ratio of highly reactive Fe ($Fe_{HR}$, Methods section) to total Fe is generally typical of anoxic conditions ($>0.38$) in all sections, with the pyrite to $Fe_{HR}$ ratio indicating ferruginous (i.e., non-sulphidic) water column conditions[28,29]. Some samples ($n=5$) fall in the equivocal ($Fe_{HR}/Fe_T=0.22$–$0.38$) or oxic ($Fe_{HR}/Fe_T<0.22$) fields (Fig. 2). However, two of these samples have Fe/Al ratios clearly indicative of anoxia (Fe/Al>0.66[30]), suggesting that

the low $Fe_{HR}/Fe_T$ ratios are likely a result of loss of unsulphidized $Fe_{HR}$ to Fe-rich clays during diagenesis[31]. Thus, our Fe-speciation data strongly support dominantly anoxic, ferruginous depositional conditions throughout most of the succession. These results are consistent with Fe-speciation data from the rest of the Neoproterozoic, suggesting ferruginous deep waters, beneath oxic shallow waters, were common throughout the period[32].

**Trace element concentrations indicate oxic weathering.** Before evaluating variations in redox sensitive TE in terms of redox conditions, it is important to demonstrate that variability is not due to changes in detrital influence alone. By normalizing TE to non-redox sensitive elements with similar geochemistry (i.e., Sc, Zr) and comparing with a detrital proxy (Al), we show that variations in detrital input, as indicated by Al, cannot fully explain these data (Fig. 3). These plots are corrected for the non-carbonate fraction. In addition, laser ablation analyses indicate that the clay fraction, and not the detrital carbonate, is the host for the TE of interest (Fig. 4). Laser analytical spots with moderate CaO and MgO, and high $K_2O$, contents compared with total major oxides generally have a higher concentration of V and U. While fine grain size and standardless data normalization[33] do not allow for quantitative conclusions based on laser ablation data alone, the analyses do indicate that the clay fraction, and not detrital carbonate, is the host of TE.

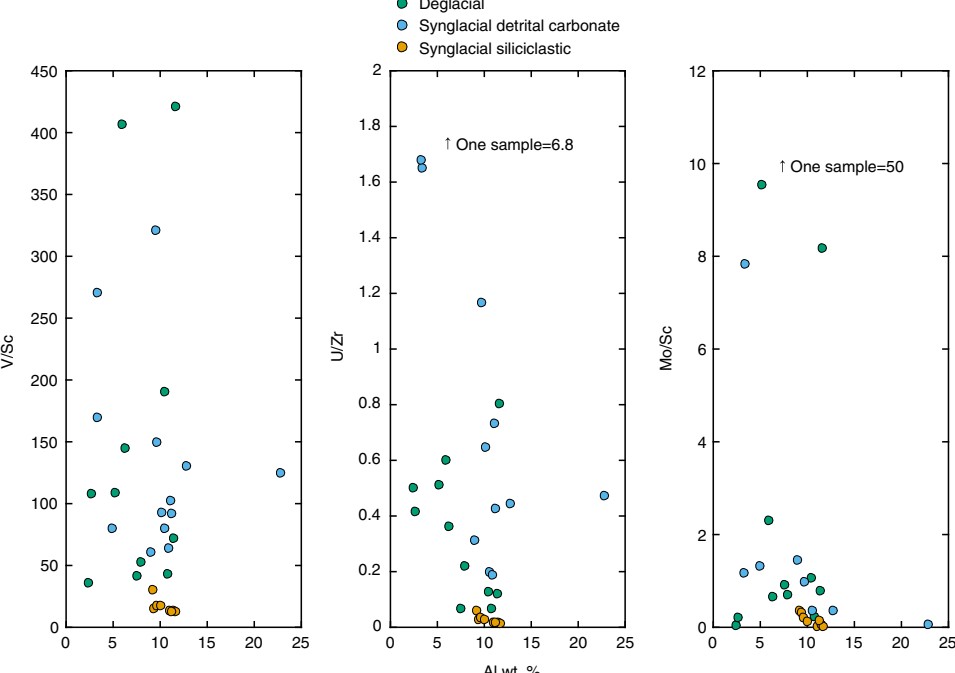

**Fig. 3** Detrital input test using Al. Whole-rock analyses of redox-sensitive trace elements of interest, normalized to geochemically similar but redox-insensitive elements and plotted against Al. If all variation seen in redox-sensitive trace elements were due to changes in detrital input alone, they would correlate to Al. Since this correlation is not seen, we suggest that variation in redox-sensitive trace elements is due to changes in water column chemistry

Trace element whole rock (WR) concentrations, reported on a carbonate-free basis, show enrichments in U, Mo, and V above Post Archean Average Shale (PAAS). Enrichments are evident in all sections, particularly in SDC and DGC. Uranium, Mo, and V concentrations, corrected for carbonate content, above PAAS requires water column anoxia to draw down the metals to the sediment[23,34], consistent with our redox reconstruction of bottom waters throughout the succession. Indeed, Mo tends to fluctuate, with concentrations which are often relatively close to PAAS. This contrasts with U in particular, which shows more significant enrichments (Figs. 2 and 4). These observations support deposition from anoxic, ferruginous bottom waters, since high concentrations of dissolved sulphide are specifically required for extensive drawdown of Mo[35,36].

The availability of the trace metals for drawdown under anoxic ocean conditions requires contemporaneous oxic weathering of the continents, since all three trace metals are mobilised as oxyanions, with modern ocean residence times of 400 Kyr for U, 50 Kyr for V and 800 Kyr for Mo;[37] these times would be shorter given lower atmospheric O$_2$, and are much less than the duration of the Marinoan glaciation (4 to 20 Myr[38]). Therefore, the delivery fluid must have been in contact with atmospheric oxygen, with limited hydrothermal influence (Supplementary Note 2). The present day lifetime of atmospheric oxygen with respect to oxic weathering is ~2 Myr[39], and would have been similar or shorter in the Neoproterozoic, when atmospheric oxygen was lower, due to the rate dependence of oxic weathering on oxygen concentration to a power of 0.5–1[40,41].

Even if oxic continental weathering were to shut off completely, a modern amount (3.5×10$^{19}$ mol) of O$_2$ would be completely exhausted in ~10 Myr given a volcanic/metamorphic reducing gas flux of 3 Tmol per year[42]. If atmospheric O$_2$ content decreases, and reducing input stays constant, the time needed to deplete atmospheric O$_2$ time will go down. Given that the residence time of atmospheric oxygen is shorter than the glaciation, a contemporaneous source of oxygen from primary productivity followed by burial of organic matter is required to maintain O$_2$ levels sufficient for oxidative weathering, as required by the TE enrichments. It is possible that a "hard" Snowball ocean may have existed for a time, but its duration would have to be much shorter than the timescale of oxic weathering (~2 Myr) in order to satisfy TE constraints, so could only have been a fraction of the glacial period. Likewise, it is possible that productivity in cryoconite holes in the ice sheet followed by transport to the ocean through moulins could account for the productivity[43]. On balance, we assert that productivity in open water is the most parsimonious explanation for our data.

**Nitrogen isotopes indicate active biologic cycling.** Our evaluation of ocean redox conditions, combined with TE evidence for oxidative continental weathering, provides crucial context for consideration of N isotope systematics. Both synglacial sections have $\delta^{15}$N values (<3‰) distinct from modern marine average sediments (~5 to 7‰), with a small but significant decrease seen up-section in SGS and an increase followed by a decrease in SGC (Fig. 2). In DGC, $\delta^{15}$N increases overall up-section, to values approaching that of modern seawater NO$_3^-$, but with more variability than either SGC or SGS. Critically, N concentrations and isotopes appear to record primary values (Supplementary Fig. 1).

Since N fixation has a very small fractionation, $\delta^{15}$N values approaching atmospheric values (assumed to be 0‰) suggest denitrification is either complete in the water column, occurs exclusively in sediments, or there is no denitrification at all. In either case, the large fractionation associated with denitrification, which leaves residual water column NO$_3^-$ isotopically heavier, will not be preserved if all NO$_3^-$ is consumed during denitrification. Similarly, sedimentary denitrification has no effect on bulk ocean $\delta^{15}$N values, as it tends to go to completion. Thus, the choices to

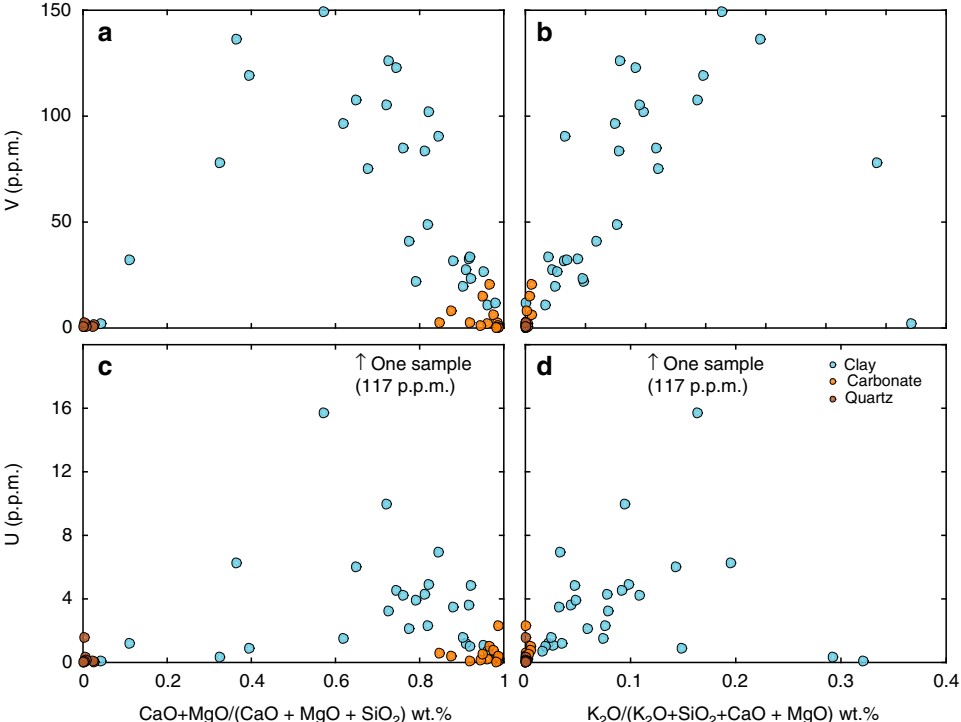

**Fig. 4** Laser ablation results showing TE are contained in clay. Laser ablation ICP-MS analyses of carbonate-, clay-, and quartz-rich locations from all sampled sections. Clay-rich fractions contain the majority of trace elements, shown clearly for V (**a**, **b**), both when plotted against proportion carbonate (CaO + MgO/(CaO + MgO + SiO₂)) or against proportion K₂O, which will be highest in clays. Although the relationship is less clear for U (**c**, **d**) it is still found predominately in clay. Since clay minerals contain most of the TEs, we suggest the TE signal directly reflects ocean water oxygen concentrations, and not a detrital overprint

explain low $\delta^{15}N$, but non-zero, values in both SGS and SGC are: a fully oxygenated water column with denitrification in the sediments only, or extensive water-column anoxia causing nearly complete, but not total, denitrification. The Fe-speciation data indicate that bottom waters were predominantly anoxic, but trace metal enrichments and non-atmospheric $\delta^{15}N$ require $O_2$ production and $NO_3^-$ availability. Thus, the low, but non-zero $\delta^{15}N$ values likely result from either a persistent or periodic shallow water oxycline.

In contrast, DGC $\delta^{15}N$ values are more enriched, and even approach modern values (which are the result of the balance between N fixation and partial denitrification in the water column) near the top of the section. This would indicate that water column denitrification is only partial at this time. That is, some $NO_3^-$ must remain in the water column to preserve the large fractionation associated with denitrification.

## Discussion

Our data from the Marinoan glaciation present a very different picture of the palaeoenvironment during low-latitude glaciation than the canonical "hard Snowball Earth" model. There is evidence of an active biosphere with oxygen production and gas exchange between the atmosphere and ocean, and contemporaneous nutrient input from the continents. There was an active nitrogen cycle, including nitrogen fixation, nitrification and denitrification. The ocean was likely dominantly ferruginous, beneath at least partially oxygenated surface waters (Fig. 5). The hard Snowball Earth model restricted ocean-atmosphere exchange and marine productivity entirely and posited an anoxic ocean throughout; our data call for that model to be rejected.

Low $\delta^{15}N$ values indicate a nitrogen cycle with nitrogen fixation, nitrification and denitrification. The balance between these is distinct from the modern balance. Nitrification occurs efficiently at $O_2$ concentrations <10 µM[44] or even nM levels[45], though denitrification would have been sufficient to process all $NO_3^-$ at such oxygen levels. Extensive, but not quantitative, denitrification is suggested, which is consistent with low oxygen levels. If removal of $NO_3^-$ produced $N_2$ or $N_2O$, N could have been limiting, though on long time scales biologic N-fixation rates are likely sufficient to mitigate limitation.

Even if there were a lower oceanic N reservoir, our dataset provides evidence for $O_2$ production and primary productivity. Evidence for N fixation itself indicates some primary production with additional productivity occurring from already fixed N. The TE data require oxidative weathering to occur throughout the glacial, which is unlikely if $O_2$ production was shut off or greatly decreased. Reaction with reducing volcanic and metamorphic gases and weathering of reduced carbon in the continental crust would exhaust atmospheric $O_2$ before the end of the glacial period without $O_2$ produced by primary production.

The robustness and flexibility of the N-cycle is also apparent when viewed in the context of the rest of the Neoproterozoic (Fig. 6), the modern ocean, and the last glacial maximum and associated deglaciation. After the Marinoan, multiple proxies indicate an increase in atmospheric $O_2$[3]. Correspondingly, $\delta^{15}N$ values increase. This trend suggests an increase in partial water-column denitrification, consistent with our interpretation of Snowball deglaciation. N-isotopic values decreased throughout the Ediacaran, perhaps in response to more quantitative denitrification, before varying widely immediately after the start of the Cambrian. Major changes to the biosphere occurred at this time, perhaps associated with changes in surface $O_2$[46].

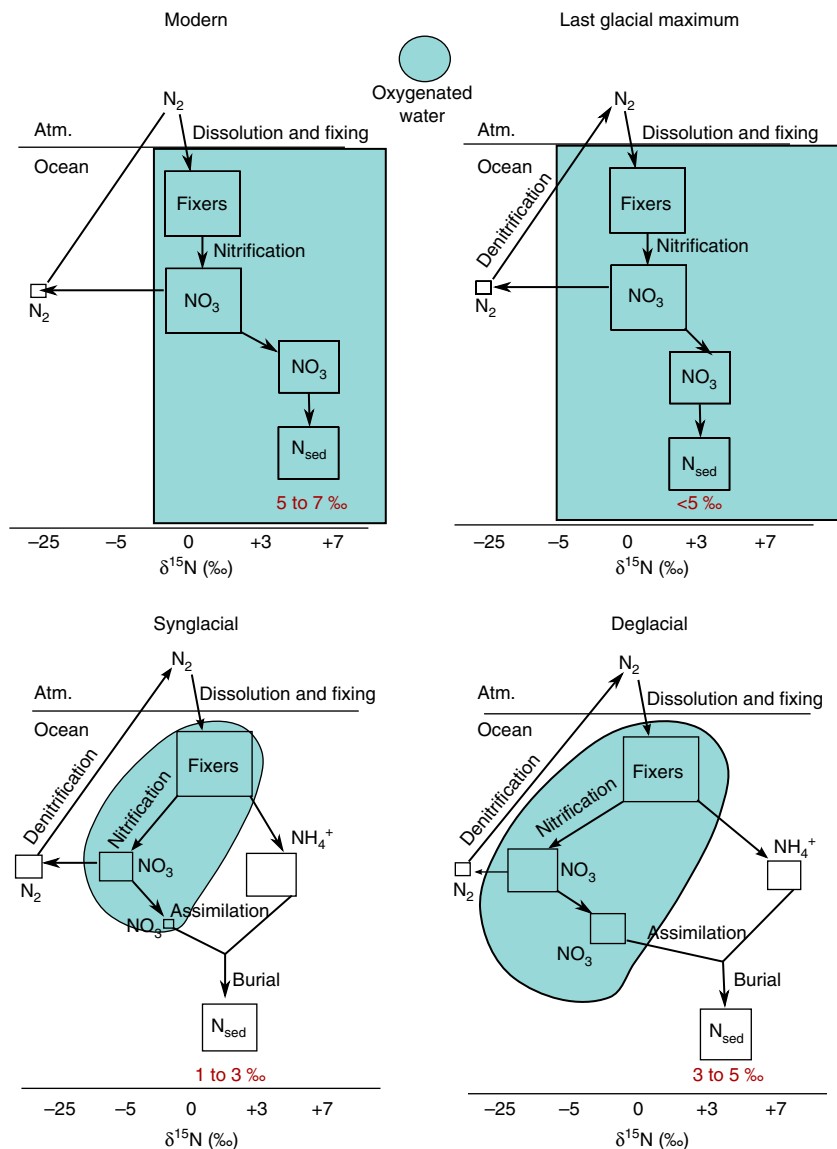

**Fig. 5** Schematic of N-cycle during disinct periods of modern and Neoproterozoic glacials. Boxes indicate size of reservoir, and position of boxes corresponds to $\delta^{15}N$ values. The modern N-cycle is characterized by partial water-column denitrification, preserving large fractionation in residual $NO_3^-$. During the LGM, extensive oxygenation limited water column denitrification, lowering $\delta^{15}N$ values globally. The Marinoan synglacial ocean had oxygenated shallow water and an extensive chemocline. Most $NO_3^-$ was denitrified, thus isotopic fractionation was not preserved. During deglaciation, higher $O_2$ allowed for larger pool of $NO_3^-$, partial water column denitrification, and an increase in $\delta^{15}N$ values approaching modern

Similarly, during the last glacial maximum, ocean waters were more pervasively oxygenated than the Holocene, and low $\delta^{15}N$ values indicate water-column denitrification was less prevalent[47]. During the subsequent deglaciation and continuing into the Holocene, N-isotopic values increase, and cover a greater range. This increasing trend reflects an increase in partial water-column denitrification and a greater range in a heterogenous ocean.

The N-cycle and the history of $O_2$ are intertwined, especially during times of great environmental change or stress. During the Neoproterozoic Marinoan glaciation, modern N-cycle processes were active, as indicated by enriched $\delta^{15}N$ values in synglacial sediments, though their relative activity was markedly different from the modern (Fig. 5). The resilience and flexibility of the N and $O_2$ cycles, as revealed by TE values, has implications not only for the survival of life on Earth, but also for life on other cosmic bodies. The presence of an active biosphere on a nearly frozen planet is indicative of life's propensity to endure.

## Methods

**Rock powder preparation**. All samples were cut and trimmed as to remove weathered exteriors. Sample chips (~1–4 cm) were then crushed into smaller pieces using a jaw crusher. Small chips were then powdered in a tungsten-carbide shatter box. Clean quartz sand was crushed in between each sample to ensure no contamination between samples occurred. The shatterbox puck and container were cleaned with deionized water and ethanol between samples.

**Nitrogen and carbon**. All N and C isotope and concentration analyses were undertaken at the University of Washington's IsoLab. Each sample was dec-arbonated prior to analyses. Approximately 1 g of sample powder was weighed into glass test tubes that had been cleaned and sterilized (baked at 500 °C overnight). Depending on the sample, between one and three 100 g tubes were prepared. To each tube, 30 mL of 6N HCl was added. Samples were then sonicated for 30 min. All tubes were then placed in a 60 °C oven overnight. The next day, samples were centrifuged to settle all undissolved material. Acid was poured off, fresh acid was added as before, and samples sat in the oven overnight. This acid refresh was repeated once more. To clean samples, all were rinsed three times with DI $H_2O$, centrifuging between each rinse. Sample powders then dried at 60 °C for

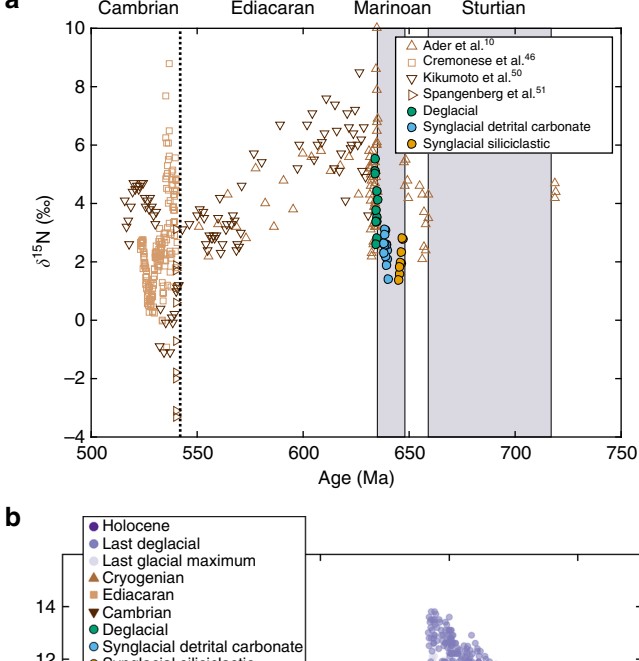

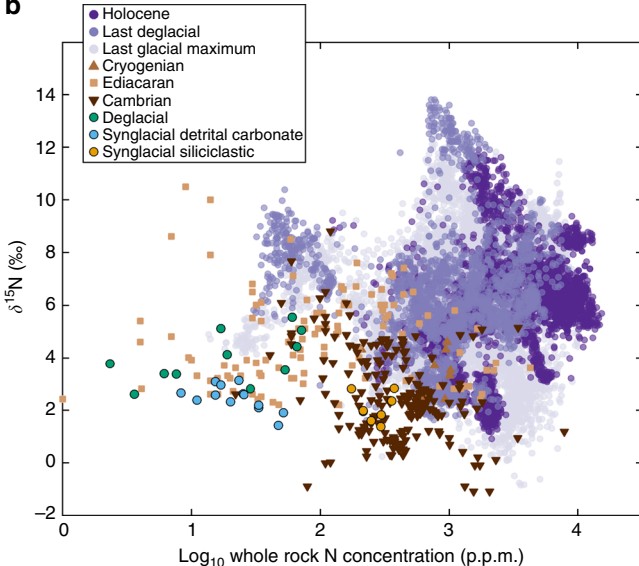

**Fig. 6** Nitrogen isotopic and whole-rock concentration. Shown are comparisons to **a** late Neoproterozoic, Cambrian, with Snowball Earth episodes marked in grey, and **b** Synglacial samples from this study are very similar to those from the Cambrian. Deglacial samples are isotopically similar to Ediacaran data as well as some modern deglacial samples. We suggest the similarity of deglacial units is due to a similar effect: increase in the amount of partial water column denitrification. The N-cycle responds dynamically to changing environmental conditions[50,51]

two days; all vials containing multiples of the same sample powder were combined and homogenized after drying.

Samples were then analysed on a Thermo-Finnigan MAT 253 coupled to a Costech Elemental Analyzer. Between 50 and 100 mg sample powder was weighed into a Sn capsule, as well as standards: two glutamic acids (GA-1 and GA-2), dried salmon (SA), and an internal rock standard (McRae Shale). All samples were flash-combusted with an excess of $O_2$ at 1000 °C in a combustion column packed with cobaltous oxide (combustion aid) and silvered cobaltous oxide (sulphur scrubber). Combustion products passed over a reduced copper column at 650 °C to reduce all N to $N_2$ and absorb excess $O_2$. Finally, sample gas passed through a magnesium perchlorate trap to absorb water and a 3 m gas chromatography column to separate $N_2$ from $CO_2$. All analyses were quantified using IsoDat software. Errors reported are standard deviations from repeated analyses.

**Trace elements**. Trace elements were measured at the University of Victoria in two ways: whole rock solutions and laser ablation. For the first, ~100 mg of samples/standards were weighed into 5 mL Teflon vials. To this, 5 mL 50% HF was added, vials were capped, and samples sat on a 170 °C hotplate for two days. Then, caps were removed, and HF was allowed to evaporate until nearly dry. Five mL of

8N nitric acid was added, vials were capped, and again samples sat on a hotplate for two days. Evaporation procedure was repeated, 5 mL of 4N nitric acid along with 0.5 mL oxalic acid was added, and samples sat on a hotplate for one day. Finally, samples were diluted to 100 mL with DI H2O in clean 100 mL bottles. Sample solutions were analysed on a Thermo X-Series II (X7) quadrupole ICP-MS. Errors were estimated from duplicate samples and standard reproducibility. We used the following standards: JLS-1, BCR-2, SY-4, SY-4, DR-N, BIR-1a, IF-G, LKSD-2, and IAEA-405.

Laser ablation was undertaken using the same mass spectrometer with a UP-213 (213 nm) solid state Nd-YAG UV laser. Beam size was set to 30 µm, and all analyses were either spots or raster lines ~100 µm in length. Samples were normalized to the sum of all major element oxides ($Na_2O$, MgO, $Al_2O_3$, $SiO_2$, $P_2O_5$, $K_2O$, CaO, MnO, and FeO) instead of an internal standard element of known concentration[33]. External standards were BCR-2g and BIR-1g. Analyses that summed to less than 90% total oxides were excluded from further discussion. In addition we tested accuracy by calculating composition of NIST 611 glass assuming it was an unknown, and based on calibration from BCR-2g and BIR-1g. Calculated trace element concentrations were within 5% of their accepted values, with most other elements within 10–15% of their accepted values (Supplementary Table 1).

TE analyses indicate the presence of oxygen in the atmosphere and shallow ocean. Samples have a major component of detrital carbonate, however, which might dilute or obfuscate TEs contained in clay minerals. We rely primarily on whole-rock (WR) analyses via solution ICP-MS for redox interpretation, as sample petrology varies at finer than laser ablation resolution. We are confident, though, that the clay mineral fraction in all samples retains the majority of TE concentration for several reasons. TE are incompatible in carbonate lattices, and should be in low concentrations. Decarbonated powders, used for N isotopic analyses, are more enriched in TE than non decarbonated WR powders. Analysis of clay-rich and clay-poor spots by laser ablation show enrichment in TE in clay-rich domains compared to carbonate grains (Fig. 4).

**Iron-based redox proxies**. The Fe speciation method targets operationally defined Fe pools, including carbonate-associated Fe ($Fe_{carb}$), ferric oxide Fe ($Fe_{ox}$), magnetite Fe ($Fe_{mag}$), and pyrite Fe ($Fe_{py}$). Extractions were performed according to well-established protocols[22,48], with subsequent analysis via atomic absorption spectroscopy (AAS) for $Fe_{carb}$, $Fe_{ox}$, and $Fe_{mag}$. $Fe_{py}$ was determined gravimetrically following chromous chloride distillation[48]. Total Fe ($Fe_T$) was determined after HF-HClO4-HNO3 dissolution via AAS. All Fe extractions gave a RSD of < 5% based on replicate analyses. Total dissolution of international sediment standards (USGS; SGR-1bl; USGS SBC-1) gave an Fe recovery of > 98%.

The sum of $Fe_{carb}$, $Fe_{ox}$, and $Fe_{mag}$, and $Fe_{py}$ defines a highly reactive ($Fe_{HR}$) pool, which is considered to represent Fe that is biogeochemically reactive during deposition and early diagenesis[22]. Modern and ancient sediments deposited from anoxic waters commonly have $Fe_{HR}/Fe_T$ ratios > 0.38, in contrast to oxic depositional conditions where ratios are generally < 0.22. Elevated $Fe_{HR}/Fe_T$ in anoxic settings arises from the additional water column formation of pyrite in euxinic (sulphidic) settings, or unsulphidized $Fe_{HR}$ minerals in ferruginous (Fe-rich) settings. Thus, the ratio of $Fe_{py}/Fe_{HR}$ distinguishes euxinic ($Fe_{py}/Fe_{HR}$> 0.7) from ferruginous ($Fe_{py}/Fe_{HR}$< 0.7) water column conditions[28]. $Fe_{HR}/Fe_T$ ratios of 0.22–0.38 are considered equivocal[29], and may occur due to the masking of water column enrichment via rapid sedimentation (e.g., during turbidite deposition), or due to transformation of unsulphidized $Fe_{HR}$ to clay minerals during diagenesis and metamorphism. This second possibility can be evaluated by considering Fe/Al ratios, since $Fe_T$ is preserved even if $Fe_{HR}$ is lost to clay minerals. In this case, normal oxic marine shales tend to have Fe/Al ratios of 0.55 ± 0.11[30], and thus Fe/Al > 0.66 is considered to provide a robust indication of water column anoxia.

Clarkson et al.[30] suggest that samples with $Fe_T$> 0.5 wt. % can generally be used to provide a robust indication of water column redox conditions[30]. Of our 32 samples, 11 have $Fe_T$< 0.5 wt. % (Supplementary Table 2), and thus these data should be considered with caution. However, we note that almost all Fe speciation data give a similar redox reconstruction, regardless of $Fe_T$ content, suggesting that all samples record a robust, anoxic ferruginous signal. In addition, however, there is a substantial component of detrital carbonate in SGC and DGC. Thus, there is the potential to add detrital Fe-carbonate minerals to these sections, which would obfuscate the Fe-speciation signal. The main Fe-carbonate phases formed under anoxic conditions are siderite and ankerite. However, the carbonate platform which served as the potential source of detrital carbonate was likely deposited under oxic conditions, and siderite and ankerite should be rare. In addition, TE data suggest oxic weathering during the Marinoan, which should serve to oxidize any detrital Fe carbonates. We thus suggest that Fe carbonate measured in all sections is dominantly primary, rather than detrital.

**Data availability**. All analytical data are available from the corresponding author (B.W.J.) via email or his website (http://benwjohnson.weebly.com). We have also placed data on EarthChem.

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

## Acknowledgements

We would like to thank Paul Hoffman for assistance in field work and selecting sampling locations and Eric Bellefroid for sampling assistance. We thank Allyson Tessin for

assistance with Fe-speciation measurements. The manuscript benefitted from comments by Rameses D'Souza and Michael Whiticar. Funding for geochemical analysis was provided via an NSERC discovery grant to CG and from the NASA Astrobiology Virtual Planetary Laboratory at the University of Washington. SWP acknowledges support from a Royal Society Wolfson Research Merit Award.

## Author contributions

B.W.J. undertook field collection of samples, as well as all geochemical analyses, and undertook manuscript preparation. S.W.P. assisted with Fe-speciation analyses and interpretation and writing of redox results. C.G. provided guidance throughout, and aided with interpretation of results with additional writing.

## Additional information

**Competing interests:** The authors declare no competing financial interests.

