## [Peer Review File · Nature Communications]

Reviewers' comments:

Reviewer #1 (Remarks to the Author):

This paper reports new synglacial isotopes (Carbon and Nitrogen) and trace element results for shale and carbonates sampled in the Neoproterozoic Snowball Earth glacial sediments. The authors interpret nitrogen isotope values to reflect an active nitrogen cycle in oxic environment during the glaciation.

I think the idea that the isotope values may reflect the presence of some nitrate in the water column during the glacial period is very interesting. However, I have a number of concerns with the present manuscript. Some of the proxies are really simplified and a direct relation between proxies and palaeoenvironmental reconstructions is too many times assumed. I developed some of my main concern below. Moreover, the text contains numerous small mistakes and typing errors. Part of them are listed below.

Major Concerns:

- I don't see that the authors have demonstrated that an oxic environment is a unique interpretation of the results provided in this paper. One of my main concerns about the true signal recorded in these sediments. It has been widely shown that diamictites are composed of an intensely mixed detrital material that cannot be directly related to the water chemistry of the glacial environment. Most of these sediment have been eroded from older layers and transported. One could say that the organic matter analyzed maybe derived from detrital organic matter which has been produced well before the glacial period.
- The trace element dataset is very interesting and the elements are interestingly chosen in order to trace redox environment. However, how the authors reconcile that Ba, U and V are highly correlated with Aluminum or Titanium contents with a redox interpretation?
- The enrichment factor is made to delete de detrital component from the authigenic phases in shales. For carbonates the EF used is more complicated. Van der Weijden (2002) showed that spurious correlations are observed in some sediments. This is why enrichment factor is usually used in Shales and not in Carbonates (where the Al contents varies highly in order of magnitude, 0,01 to 27% in the samples analyzed of this study).
- The authors get rid of the diagenesis very fast for example they claim that during diagenesis N could be trapped in the clays. Thus what happened when the lithology change to carbonates? There should be less nitrogen trapped in the sediment and thus small fractionation observed. Another concern is about the interpretation of these positive values which could be explain too with partial ammonium nitrification in an anoxic environment.
- There is a number of grammatical errors, below are few examples.

More detailed comments:

Generally, the authors should use the larger space and high number of citations authorized by Nature Communications.

-Line 4: Homogenized "Snowball" or "snowball"

-Line 15-17: I would agree with this statement. The authors must demonstrate that these sediments are non detritic organic matter and that the carbonate did not precipitated before the glacial event.

-Line 26: Add a "may" before "record". Ader et al. stipulated, on the contrary, that there are at least 6 steps to reconcile before obtaining a past biological activity from the nitrogen isotope signal.

-Line 27-28, Not only. Secondary fluids from weathering or secondary hydrothermal activity should be envisaged.

-Line 32: the reference cited in here does not show -2 or +2 per mil. Instead it stipulated that the fixation could produce d15N of -5, -6 per mil. Please change the reference.

-Page 2, in the formula of stable isotopes. Change: "the d15N standard is N2 in Air" by "the standard used for d15N values is N2 in air".

-Line 43: Add an appropriate reference after "this process has little isotopic fractionation"

-Line 50: Does the authors mean "total" by "extensive"?

-Line 52: The nitrogen isotopic cycle story is more complex. Could the author develop too the case of d15N positive values in fully anoxic or stratified ocean?

- Line 60: Enriched compared to what? What is the baseline for detrital inputs? Is that the PAAS ? In such cases authors should check by cross plotting their elements with Aluminium content.

-Line 66: Replace Ghuab by Ghaub

-Line 67: Replace formation by Formation (and homogenize it in the main text)

-Line 87: Replace Keliberg by Keilberg.

Again, I clearly understand the authors have made an effort for sampling the part of the section which may have record the water column signal. However, their data and the sedimentological description should be more convincing.

-Line 88: The authors seem to use the well bedded sediment as a proof of the synglacial formation. However, it is clearly stated that this sediments are detrital sediments. Could the organic matter comes from older layers (see Johnston et al. 2012 in Nature) ? Could the authors develop and explore this this statement in more detail?

-Line 91: Majuscule et "synglacial"

-Line 92-94: The paper would greatly benefit of a figure showing the results with the stratigraphic column.

-Line 95-100: This whole paragraph should be more developed and extended to early diagenesis processes.

-Line 104: Could the authors plot the K vs N content. Which may help to demonstrate that the N is contained in the Clays.

-Line 108: This could too mean no denitrification at all.

-Line 124-125: Becarefull with the residence time estimation. The values given in the paper are the modern ones after a long oxic environment allowing the accumulation of these trace metals in the ocean. In past environment with less oxygen these values would decrease rapidly.

-Line 133: I disagree with this statement. I do not see any correlation. Could the author support

their statement by a cross-plot? More over a clear positive covariation is observed between Ba and Al, U and Al and V and Al. I would rather interpret part of these data as reflecting variations in the detrital inputs material.

-Line 152: Again the detrital inputs is a problem. I recommend the author to use Mo/TOC and U/TOC ratios for trying to decipher between oxic and anoxic.

-Line 151-152: there is here to me a paradox if trace element indicates anoxic water column and $\delta^{15}N$ oxic water column.

-Line 176-178: Could the author support their hypothesis? In the model published, because of the large atmospheric reservoir, it is very hard to have an anoxic atmosphere even during a snowball Earth scenario.

Figure 1:

The Chuos Formation is written Chous in the Figure.

Check the majuscule in the legends and the figure itself.

Could the authors indicate where the cap carbonate samples come from on the Figure 1c.

Figure 3:

Can the authors add a graph for Aluminium contents or Titanium.

Reviewer #2 (Remarks to the Author):

Johnson and Goldblatt present new nitrogen isotope and trace element data for Neoproterozoic syn-glacial strata of the Ghaub Formation, deposited during the Marinoan 'Snowball Earth' glaciation. The authors interpret the geochemical data to reflect an active N redox cycle, with nitrogen fixation, nitrification, and partial denitrification, vigorous local biological productivity (based on sedimentary Ba enrichments), oxidative weathering of redox-sensitive trace elements, and redox heterogeneity in the ocean interior.

Generally, I think the N isotope data represent an important contribution, given no previous syn-glacial data. These data provide an interesting contrast to later Ediacaran data, and will be of broad interest. However, I am skeptical of the trace element interpretations, and feel that the overall narrative is structured to counter what is essentially a straw man at this point -- the 'hard' snowball hypothesis. In other words, I don't think that many (any?) people would currently go to the mat for the end-member 'hard snowball' model. I may be mis-reading the state of play, but if so I doubt I would be the only one, so I think the authors need to cite some recent literature that still pushes this model. Broadly, I think the data should be published, but would recommend major revisions to address the issues discussed below.

First, the authors use sedimentary Ba content as a paleoproductivity proxy, but the rationale behind this, the caveats involved, and the potential uncertainties are not adequately discussed. The text essentially bases its entire argument in this regard on a single clause, "Barium is a productivity proxy" (Line 132), citing a single paper from 1992. However, there are significant caveats associated with application of the Ba proxy -- uncertain sources (e.g., Griffith & Paytan, *Sedimentology*, 2012), provenance effects with direct ramifications for calculated EF values (Klump et al., *Marine Geology*, 2000), the potential for differential remobilization in suboxic sediments or reducing basins (e.g., McManus et al., *GCA*, 1998), and phase associations that are often complex (Gonneea & Paytan, *Marine Chemistry*, 2006). None of this is addressed or discussed in the manuscript.

More broadly, the abstract and text make statements such as: "Sedimentary barium enrichment

further indicates biological productivity". This may be fair enough (subject to the caveats above), but is a somewhat vague statement. I am not familiar with any literature suggesting that the oceans were sterilized during the Neoproterozoic glacials, so it stands to reason that there would be regions that feature "biological productivity". I think in order for this to be meaningful the authors really need to address this quantitatively -- e.g., how much productivity? What does this mean for different models of glaciation?

It is odd to me that the U and V data are shown as both bulk-rock enrichments and calculated EF values, but the Mo data are not. This is important, as these rocks are extremely organic-lean (mostly well below ~0.1wt%), and if the authors are to break the degeneracy that they (correctly) highlight in N isotope data they need to have some confidence that the sedimentary U/V/Mo systematics are telling them something about the redox state of the overlying water column. They have simply not made this case convincingly, in my view, and detailing the bulk-rock Mo values would help in this regard. Enrichment factors can be quite misleading, particularly given that most of these lithofacies are not the fine-grained, organic-rich depositional systems that these redox proxies are most extensively calibrated for.

Along a similar line, the authors present laser ablation data which are used to argue: "Since clay minerals contain most of the TEs, we suggest the TE signal directly reflects ocean water oxygen concentrations" (caption to Figure 3). There are two problems here. First, it is not clear to me how clay minerals "containing" the TE signals helps the arguments presented in the paper -- if read at face value, this would suggest that the metals are predominantly detrital, and thus tell you very little about overlying water chemistry. I assume this is not really what the authors mean -- e.g., what they really mean is that the TEs are hosted in spots that appear dominated by clay mineral phases, but there is very little detailed discussion of the laser ablation work in the main text or methods. The authors quote a 30um spot size, and refer to ~100um raster lines, but it is never made clear how each of these were applied, or what is even meant by 'clay minerals' in the in-situ context. Further, neither the V or U enrichments shown are dramatically elevated, and if the Mo content is similarly low it is very likely that much of the metal inventory in these sediments is associated with the detrital load. This would be fully consistent with the authors' contention that the metals are largely hosted with clay minerals, but would strongly mitigate any proxy information for the chemistry of the overlying water column.

I take the authors' point that continued oxidative weathering and trace element delivery may require a sustained O₂ source given the duration of the glaciation. However, I think this argument would be greatly strengthened by a quantitative consideration of how crustal weathering sinks would be expected to scale back during a large-scale glaciation. Just a back-of-the-envelope would do. For example, assuming that crustal weathering ceases entirely, and volcanic reductant input is on the order of, say, ~3 Tmol/y (in O₂ equivalents), at 1PAL the long-term residence time of O₂ in the atmosphere would be on the order of 10 Myr. If atmospheric pO₂ was less, the residence time would go down. So, I don't disagree with the authors' argument here, but I think that it won't be transparently obvious for many readers that the strong source need be required and that a simple calculation will make this more accessible and compelling.

I'm not sure that Figure 5 is really necessary. It doesn't really contribute any essential or novel information that the text and other figures don't address.

The final paragraph seems cursory and vague. For example, "Even during periods of global glaciations, organisms procured nutrients and survived" (Lines 192-193) is bordering on tautological given phylogenetic constraints. I think I understand what the authors are going for here, but it needs to be developed a little further and clarified.

"Ghuab" is mis-spelled in Line 153.

Reviewer #3 (Remarks to the Author):

Review of Marine primary productivity and oxygen production during Snowball Earth
By Ben Johnson and Colin Goldblatt

This manuscript presents a chemostratigraphic study of Neoproterozoic units in SW Africa deposited during and following the Marinoan glaciation. The problem investigated is whether there is evidence for oxygen production by marine photosynthesizers during this glacial event, which might serve as evidence for less-than-total glaciation of the Earth's oceans (the "hard snowball Earth" model).

The data produced are of a high-quality, the manuscript is generally well-written and -illustrated, and the interpretations are mostly reasonable. Overall, I think that this study may be publishable following moderate revision.

Major points:

Conclusions--I am in agreement with the authors' main conclusion that their geochemical results suggest that oxygen production occurred during much of the Marinoan glaciation, and that therefore, a "hard snowball Earth" did not exist for the duration of the glaciation (4-20 Myr). On the other hand, I would like the authors to consider whether a "hard snowball Earth" might have existed for a fraction of the Marinoan glaciation, e.g., during the interval of the SGS, followed by a longer interval of less-complete glaciation. Are their data permissive of this possibility?

Redox interpretations--The authors use trace metal concentrations (or enrichment factors) to evaluate paleoredox conditions in the study units. This is, broadly speaking, OK, and certainly the high EFs of some units suggest strongly reducing conditions. The matter is less clear for the SGS unit, which shows low EFs. These low EFs might indicate oxic watermass conditions, or possibly a reservoir effect (local drawdown of aqueous trace metals) or limited oxic subaerial weathering of trace metals. The uncertainty here would be best resolved through Fe speciation analysis, which would be highly desirable for these units.

Ba proxy--It is unclear whether the use of Ba as a paleoproductivity proxy is justified. In the modern open ocean, sediment Ba concentrations reflect productivity because (i) the concentrations of Ba in seawater and sulfate on the surface of decaying organic particles exceed the solubility product of barite, (ii) there is ample time for barite to accumulate during the ~5000 m sinking of organic particles to the deep-ocean floor, and (iii) barite remains thermodynamically stable under oxic conditions. Potential problems: (1) The study units were presumably not deposited in the open ocean (the text states only "deep water" on lines 76-77, but these are epicratonic units, so water depths were probably no more than a few hundred meters), limiting time for Ba uptake. (2) If the water column was anoxic, then barite would have been thermodynamically unstable, with a tendency to dissolve. (3) In many regions globally, the Neoproterozoic was a time interval of intense hydrothermal activity, and the Ba might have been hydrothermally sourced. The authors could address these uncertainties first by determining what phase the Ba is present in (barite or something else?), and second by investigating whether the elemental chemistry of the study units provides any indication of hydrothermal inputs (e.g., Fe/Mn ratios).

Alteration of N isotopes--On lines 102-104 it is stated: "If N were volatilized and lost with progressive metamorphism, concentration and $\delta^{15}\text{N}$ would be negatively correlated, which is not observed (Fig. 2)." In fact, there appears to be a modest negative correlation between [N] and $\delta^{15}\text{N}$ in Figure 2A. The R2 might be only 0.2-0.3 but given the large number of samples present, this may well be a significant correlation. What is the R2 value and significance level for the relationship in Figure 2A? Doesn't this in fact suggest that the N-isotope compositions of the study samples may have been affected by deep burial/metamorphic processes?

Modern versus Neoproterozoic values--The authors need to exercise some caution in citing modern ocean values for various parameters. These values are not necessarily valid for the Neoproterozoic. For example, the seawater residence times for trace metals cited on lines 123-124.

This issue also arises on Lines 144-145: "Uranium is generally more enriched than Mo, so the water is not euxinic." This statement depends on seawater having the modern ratio of Mo to U. If,

however, the Mo/U ratio of Neoproterozoic seawater were significantly lower than the modern value, then this statement would be incorrect. Again, greater caution is required.

Minor issues:

There are some grammatically incorrect sentences, e.g., lines 127-128, 132-133, 146-147.

"Upsection" is a single word (given incorrectly on lines 92-93). "Synglacial" can be written without a hyphen. So can "nearshore".

Line 93: "modern ocean bulk $\delta^{15}\text{N}$ ". It would be more accurate to write "modern seawater nitrate $\delta^{15}\text{N}$ ".

"N-fixing" (line 106 and elsewhere) should be changed to "N fixation".

Line 120: "between oxic, allowing TE to accumulate". This sentence needs to be refined to "... accumulate in the water column" (as opposed to the sediment). The present wording is ambiguous.

The authors define three stratigraphic intervals: synglacial siliciclastic (SGS), synglacial carbonate (SGC), and deglacial carbonate (DGC). These intervals are shown and labeled in Figure 4 (although the acronyms are not shown here, but they could be). These intervals probably correspond in some manner to the three groups of sample points shown in Figure 1, but those points are not labeled. The ranges of the SGS, SGC, and DGC units need to be shown clearly in Figure 1.

Figure 3--are the relationships shown here important? The four graphs just show that trace metals are enriched in the shaly samples (high K₂O) and depleted in the carbonate-rich samples (high CaO+MgO); there is also a small group of samples located at the origin of all four graphs (and thus probably cherts) that contain near-zero trace metal concentrations. These are completely unsurprising findings, and they could probably be stated in the text without the need for a figure (the figure could be transferred to an SI file, for example).

Figure 4--Why are TOC and $\delta^{13}\text{C}$ profiles not included in this figure? These data are integral to the study's interpretations, so their secular variation should be illustrated in one of the figures.

Thank you to all three reviewers and to the editor. We found the feedback very constructive and helpful to the manuscript. Our responses are shown below in black text, with original reviewer comments in grey.

Reviewers' comments:

Reviewer #1 (Remarks to the Author):

This paper reports new synglacial isotopes (Carbon and Nitrogen) and trace element results for shale and carbonates sampled in the Neoproterozoic Snowball Earth glacial sediments. The authors interpret nitrogen isotope values to reflect an active nitrogen cycle in oxic environment during the glaciation.

I think the idea that the isotope values may reflect the presence of some nitrate in the water column during the glacial period is very interesting. However, I have a number of concerns with the present manuscript. Some of the proxies are really simplified and a direct relation between proxies and palaeoenvironmental reconstructions is too many times assumed. I developed some of my main concern below. Moreover, the text contains numerous small mistakes and typing errors. Part of them are listed below.

Major Concerns:

- I don't see that the authors have demonstrated that an oxic environment is a unique interpretation of the results provided in this paper. One of my main concerns about the true signal recorded in these sediments. It has been widely shown that diamictites are composed of an intensely mixed detrital material that cannot be directly related to the water chemistry of the glacial environment. Most of these sediment have been eroded from older layers and transported. One could say that the organic matter analyzed maybe derived from detrital organic matter which has been produced well before the glacial period.

We accept that diamictites, and other glacial facies, have a high proportion of detrital material sourced from a a variety of lithologies. When have added both figures and discussion to address this issue. Please see below for discussion concerning detrital influence on redox-sensitive trace elements. Concerning N, we suggest that N is contained mainly in clay minerals, as indicated by a N vs Rb correlation (Supplementary Fig. 9). Nitrogen is known to substitute for K in clay minerals, and Rb will as well. We did not analyze K in our WR dissolution analyses. Nitrogen has been shown to substitute into clay minerals after being deposited from the water column, and should provide a primary record of water column $\delta^{15}\text{N}$ character.

- The trace element dataset is very interesting and the elements are interestingly chosen in order to trace redox environment. However, how the authors reconcile that Ba, U and V are highly correlated with Aluminum or Titanium contents with a redox interpretation?

While bulk Al and elements of interest are correlated, we suggest that this is likely because Al and trace elements are all more enriched in the non-carbonate fraction of the samples. This is evident in a plot of V or Al vs residue (i.e., material left after decarbonation).

We have included new figures where we substantially address this concern. In these plots, we normalize trace elements (V, U, Mo, plus Ba) to a non-redox sensitive trace element (Sc for V, Mo and Ba and Zr for U) that behaves similarly to trace elements of interest. A plot, then, of this ratio (e.g., V/Sc) against Al shows weak to no correlation. If changes in detrital input alone were responsible for

changes in all element concentrations, we would expect the V/Sc, Mo/Sc, U/Zr, and Ba/Sc ratios to vary linearly with Al. They do not, thus we suggest that changes in redox-sensitive trace elements and Ba are due to changes in water column chemistry and productivity, respectively.

In addition, we note that the organic content of these samples are quite low. We suspect that the N is primarily contained within clay minerals, as evidenced by the loose correlation between N and Rb. Since organic matter is low, we interpret that there is little detrital organic matter.

We have used laser ablation analyses to demonstrate that the carbonate fraction does not contribute to trace element whole rock analyses. Thus, by presenting data by correcting for carbonate fraction, we suggest that the detrital influence is small.

- The enrichment factor is made to delete the detrital component from the authigenic phases in shales. For carbonates the EF used is more complicated. Van der Weijden (2002) showed that spurious correlations are observed in some sediments. This is why enrichment factor is usually used in Shales and not in Carbonates (where the Al contents varies highly in order of magnitude, 0,01 to 27% in the samples analyzed of this study).

Thank you for highlighting this reference. We presented in our original submission EFs that are for whole rock analyses that are corrected for carbonate content. This approach then "scales up" clay analyses. That is, we are essentially treating the non-carbonate fraction as a pseudo-whole rock proxy. We did not rely on correlations between elements, rather we relied on EFs of individual elements for redox interpretations.

- The authors get rid of the diagenesis very fast for example they claim that during diagenesis N could be trapped in the clays. Thus what happened when the lithology change to carbonates? There should be less nitrogen trapped in the sediment and thus small fractionation observed. Another concern is about the interpretation of these positive values which could be explain too with partial ammonium nitrification in an anoxic environment.

We note that all units are primarily detrital carbonates, but we focus our analyses on the non-carbonate fraction. You are correct in noting that given a lower non-carbonate fraction (e.g., section J1408) the overall N concentrations are lower.

Your point regarding the possibility of partial ammonium nitrification is well noticed. We agree that there must have been partial ammonium nitrification during the glaciation, but then our interpretation assumes that most of this nitrate is denitrified, slightly enriching $\delta^{15}\text{N}$ values of residual dissolved N overall. Indeed, nitrification occurs at very low oxygen levels (nM O_2 , Füssel et al., 2012 and Thamdrup et al., 2012) and was likely occurring even if conditions were nearly anoxic.

Even if there was only partial nitrification (as is our suggestion), remaining N-isotopic composition of non-nitrified N would be enriched regardless of whether nitrate is denitrified or not. So we agree with your suggestion that there was partial nitrification, but then we suggest our trace element data indicating some oxygenated waters is consistent with near-quantitative denitrification.

- There is a number of grammatical errors, below are few examples.

More detailed comments:

Generally, the authors should use the larger space and high number of citations authorized by Nature Communications.

-Line 4: Homogenized "Snowball" or "snowball"

Thank you for noticing, we have changed all to be capitalized.

-Line 15-17: I would agree with this statement. The authors must demonstrate that these sediments are non detritic organic matter and that the carbonate did not precipitated before the glacial event.

We have reworded several sentences to clarify that we do not think carbonates carry information on the redox state of the Snowball ocean. Instead, we rely on clay fraction analyses for our interpretation. In addition, these units are poor in organics, with N likely found as NH₄ in clay fraction.

-Line 26: Add a "may" before "record". Ader et al. stipulated, on the contrary, that there are at least 6 steps to reconcile before obtaining a past biological activity from the nitrogen isotope signal.

We have added "may", this is an important qualification to state.

-Line 27-28, Not only. Secondary fluids from weathering or secondary hydrothermal activity should be envisaged.

We have changed this paragraph to include "Post-depositional fluid alteration could also affect values."

-Line 32: the reference cited in here does not show -2 or +2 per mil. Instead it stipulated that the fixation could produce $\delta^{15}\text{N}$ of -5, -6 per mil. Please change the reference.

We have added two new references: Hoerning and Ford, 1960 and Zerkle et al., 2008, which are more appropriate and show minimal fractionation under modern conditions with Mo-based nitrogenase.

-Page 2, in the formula of stable isotopes. Change: "the $\delta^{15}\text{N}$ standard is N₂ in Air" by "the standard used for $\delta^{15}\text{N}$ values is N₂ in air".

Changed.

-Line 43: Add an appropriate reference after "this process has little isotopic fractionation"

We have added Frenndenthal et al., 2001 and also added that isotopic fractionation is small especially under anoxic sedimentary conditions.

-Line 50: Does the authors mean "total" by "extensive"?

Changed to "quantitative"

-Line 52: The nitrogen isotopic cycle story is more complex. Could the author develop too the case of $\delta^{15}\text{N}$ positive values in fully anoxic or stratified ocean? There are indeed complications given a completely anoxic ocean. We explore this more fully in lines 57-64, but still suggest that combining N with redox-sensitive trace element data is a useful, and indeed necessary tool in our analysis.

We now say in lines 57-64:

A complication could arise if there is a fully anoxic ocean with ample NH₄ . Since there is a fractionation associated with NH₄ -assimilation, one would expect biomass in a N-replete ocean to be negative [17]. Modern sediments generally reflect biologic cycling in the overlying water column [15], thus

negative $\delta^{15}\text{N}$ values could record such an anoxic, but N-replete environment. If there is a deficit in dissolved NH_4 , however, the fractionation from assimilation would not be preserved, as all available N would likely be assimilated into biomass. Complete assimilation of NH_4 in an anoxic should result in biomass that is isotopically equal to fixed N. Again, we recognize the need for a redox proxy to assist in interpretation of N-isotopic data.

- Line 60: Enriched compared to what? What is the baseline for detrital inputs? Is that the PAAS? In such cases authors should check by cross plotting their elements with Aluminium content.

We have made a new figure to address issues of detrital influence. We show elements of interest (V, U, Mo, and Ba) plotted against Al. Elements of interest are normalized to either Sc (V, Mo, Ba) or Zr (U), as Sc and Zr are geochemically similar to V

-Line 66: Replace Ghuab by Ghaub
Changed, thank you.

-Line 67: Replace formation by Formation (and homogenize it in the main text)
Changed, thank you.

-Line 87: Replace Keliberg by Keilberg.
Changed, thank you

Again, I clearly understand the authors have made an effort for sampling the part of the section which may have record the water column signal. However, their data and the sedimentological description should be more convincing.

-Line 88: The authors seem to use the well bedded sediment as a proof of the synglacial formation. However, it is clearly stated that this sediments are detrital sediments. Could the organic matter comes from older layers (see Johnston et al. 2012 in Nature)? Could the authors develop and explore this this statement in more detail?

It was our intent to suggest that the stratigraphic relationship between the finely laminated units and the units with dropstones indicates their glacial origin. Specifically, the sampled units are interbedded with dropstone bearing units, with the latter clearly from glacial deposition.

We have also suggested more clearly that there was little detrital influence on N. We say, on lines 102-109:

In all units, there is the potential that any measured biogeochemical data samples reworked,

detrital material. Any such material could mask information about the water column during deposition. However, modern N in most marine settings mostly comes from sinking organic material sourced from primary production. In addition, previous work suggests the source of the majority of Ghuab sediments was the Ombaatjie Fm., representing a near-shore carbonate platform [19], The Ombaatjie Fm. is relatively organic-poor [22]; thus, we suggest that detrital organic input is minimal in the sampled sections.

-Line 91: Majuscule et "synglacial"
Fixed, thank you.

-Line 92-94: The paper would greatly benefit of a figure showing the results with the stratigraphic column.

Thank you for this suggestion. As we sampled units of similar lithology, and due to analytical constraints were not able to produce fine-scale geochemical data. That is, each hand sample has many fine (<1cm) laminations, but we were not able to analyze each lamination. Our data is only representative of bulk samples, and presentation next to the stratigraphic column does not significantly add to our data and interpretations.

-Line 95-100: This whole paragraph should be more developed and extended to early diagenesis processes.

We present a supplementary figure of Cs vs. Zr. Both elements are incompatible, while Cs is very fluid-mobile and Zr is very fluid immobile. Linear correlation between these two suggest no large-scale post-depositional fluid alteration has occurred. This suggests that N values are likely also unaffected by later fluid flow.

-Line 104: Could the authors plot the K vs N content. Which may help to demonstrate that the N is contained in the Clays.

We did not measure K in our whole rock digestions, but we present a figure of N vs Rb instead. There is a positive correlation between the two, supporting our hypothesis of clays as the host of N.

-Line 108: This could too mean no denitrification at all.

Perhaps, though even in high oxygen waters, at a certain depth in the sediment one would expect to find low-O₂ conditions, and denitrification, given a supply of organic matter to respire. We have added a statement acknowledging the possibility of no denitrification.

-Line 124-125: Be carefull with the residence time estimation. The values given in the paper are the modern ones after a long oxic environment allowing the accumulation of these trace metals in the ocean. In past environment with less oxygen these values would decrease rapidly.

Agreed, we have clarified that these residence times are for the modern ocean. If anything, shorter residence times strengthens our argument for continued oxic weathering, as short residence times of TEs in the ocean decrease the overall pool dissolved in the ocean and decrease their likelihood of being found in sediments.

-Line 133: I disagree with this statement. I do not see any correlation. Could the author support their statement by a cross-plot? More over a clear positive covariation is observed between Ba and Al, U and Al and V and Al. I would rather interpret part of these data as reflecting variations in the detrital inputs material.

We now show a plot of Ba/Sc vs Al all corrected for non-carbonate fraction. If variations in all three elements were controlled by changes in detrital input alone, this relationship would be a linear correlation. Since the relationship is not a linear correlation, we suggest there must be at least one other factor controlling element variation. One possibility is that changes in Ba are due to changes in biologic productivity, though there are uncertainties in this proxy under anoxic conditions. These uncertainties are now highlighted in this paragraph.

-Line 152: Again the detrital inputs is a problem. I recommend the author to use Mo/TOC and U/TOC ratios for trying to decipher between oxic and anoxic.

We suggest that TOC values are altered in these samples, and are likely unreliable. As an alternative, we have plotted U, V, and U normalised to Sc plotted against Al for carbonate-corrected samples as a way to assess detrital influence.

-Line 151-152: there is here to me a paradox if trace element indicates anoxic water column and d15N oxic water column.

This was not made clear in our original submission, thank you for noticing. We have clarified this paradox by indication that the water column must have varied between

oxic/sub-oxic and anoxic/euxinic, lines 200-203:

We suggest, then, that both the higher $\delta^{15}\text{N}$ values and EFs indicate a water column that varies between oxic to sub-oxic, allowing partial water column denitrification, and anoxic to euxinic, when TEs are scavenged from the water column and into the sediment.

-Line 176-178: Could the author support their hypothesis? In the model published, because of the large atmospheric reservoir, it is very hard to have an anoxic atmosphere even during a snowball Earth scenario.

We present now a back of the envelope style calculation, as suggested by Reviewer #2:

Even if oxic continental weathering were to shut off completely, a modern amount (3.5×10^{19} mol) of O_2 would be completely exhausted in ~ 10 Myr given a volcanic/metamorphic reducing gas flux of 3 Tmol yr^{-1} [31]. If atmospheric O_2 goes down, and reducing input stays constant, the time needed to deplete atmospheric O_2 time will go down. Given that the residence time of atmospheric oxygen is shorter than the glaciation, a contemporaneous source of oxygen from primary productivity followed by burial of organic matter is required to maintain O_2 levels sufficient for oxidative weathering, as required by TE concentrations.

Figure 1:

The Chuos Formation is written Chous in the Figure.

Check the majuscule in the legends and the figure itself.

Could the authors indicate where the cap carbonate samples come from on the Figure 1c.

Typo corrected, thank you. We have also corrected capitalization. We actually did not collect any samples from the cap carbonate. We have reworded the geologic setting and sample description section to add clarity.

Figure 3:

Can the authors add a graph for Aluminium contents or Titanium.

Yes. This is shown in Figure 5.

Reviewer #2 (Remarks to the Author):

Johnson and Goldblatt present new nitrogen isotope and trace element data for Neoproterozoic syn-glacial strata of the Ghaub Formation, deposited during the Marinoan 'Snowball Earth' glaciation. The authors interpret the geochemical data to reflect an active N redox cycle, with nitrogen fixation, nitrification, and partial denitrification, vigorous local biological productivity (based on sedimentary Ba enrichments), oxidative weathering of redox-sensitive trace elements, and redox heterogeneity in the ocean interior.

Generally, I think the N isotope data represent an important contribution, given no previous syn-glacial data. These data provide an interesting contrast to later Ediacaran data, and will be of broad interest. However, I am skeptical of the trace element interpretations, and feel that the overall narrative is structured to counter what is essentially a straw man at this point -- the 'hard' snowball hypothesis. In other words, I don't think that many (any?) people would currently go to the mat for the end-member 'hard snowball' model. I may be mis-reading the state of play, but if so I doubt I would be the only one, so I think the authors need to cite some recent literature that still pushes this model. Broadly, I think the data should be published, but would recommend major revisions to address the issues discussed below.

First, the authors use sedimentary Ba content as a paleoproductivity proxy, but the rationale behind this, the caveats involved, and the potential uncertainties are

not adequately discussed. The text essentially bases its entire argument in this regard on a single clause, "Barium is a productivity proxy" (Line 132), citing a single paper from 1992. However, there are significant caveats associated with application of the Ba proxy -- uncertain sources (e.g., Griffith & Paytan, *Sedimentology*, 2012), provenance effects with direct ramifications for calculated EF values (Klump et al., *Marine Geology*, 2000), the potential for differential remobilization in suboxic sediments or reducing basins (e.g., McManus et al., *GCA*, 1998), and phase associations that are often complex (Gonneea & Paytan, *Marine Chemistry*, 2006). None of this is addressed or discussed in the manuscript.

Thank you for making us aware of this, we were unaware of these issues in the original submission. In our revised submission, we are less reliant on the Ba data as a robust proxy for primary productivity. Specifically, in lines 222–231:

Barium is a productivity proxy in the modern ocean [43]. Barium released during organic matter degradation bonds with sulphate to form barite, which is preserved in oxic sediments [44]. High Ba EFs (Fig. 6) in all sections may indicate active biologic ecosystems during Snowball ocean was productive. This proxy, however, is sensitive to oxygen concentrations, as barite tends to dissolve in anoxic waters, since SO_2^- is undersaturated. While the Ba-proxy is also sensitive to changes in detrital influence [45], we suggest that Ba concentrations are not controlled by detrital influence alone (Fig. 5), as Ba/Sc ratios do not correlate with Al content when corrected for non-carbonate fraction. It is thus possible, though speculative, that there is some biologic control over Ba abundances. The rough correlation between Ba EF and $\delta^{15}\text{N}$ in SGC suggests that in SGC the N-cycle and productivity are linked.

More broadly, the abstract and text make statements such as: "Sedimentary barium enrichment further indicates biological productivity". This may be fair enough (subject to the caveats above), but is a somewhat vague statement. I am not familiar with any literature suggesting that the oceans were sterilized during the Neoproterozoic glacials, so it stands to reason that there would be regions that feature "biological productivity". I think in order for this to be meaningful the authors really need to address this quantitatively -- e.g., how much productivity? What does this mean for different models of glaciation?

We agree that the majority of the Snowball Earth community rejects the canonical "hard Snowball" model. There do remain several adherents to this model, however, and we suggest that framing the paper in terms of different models of the Snowball ocean is a useful approach.

It is odd to me that the U and V data are shown as both bulk-rock enrichments and calculated EF values, but the Mo data are not. This is important, as these rocks are extremely organic-lean (mostly well below ~0.1wt%), and if the authors are to break the degeneracy that they (correctly) highlight in N isotope data they need to have some confidence that the sedimentary U/V/Mo systematics are telling them something about the redox state of the overlying water column. They have simply not made this case convincingly, in my view, and detailing the bulk-rock Mo values would help in this regard. Enrichment factors can be quite misleading, particularly given that most of these lithofacies are not the fine-grained, organic-rich depositional systems that these redox proxies are most extensively calibrated for.

We now show all TE data as whole rock (WR), normalized to Zn (a proxy for organic C), and as enrichment factors, with all corrected for non-carbonate fraction in

Figure 6. Based on the similarity between the WR and Zn-normalised plots, we suggest that redox-sensitive TEs were originally associated with organic matter, as is observed in organic-rich shales. Thus, the use of EFs are instructive, even though they were originally intended for organic-rich shales.

Along a similar line, the authors present laser ablation data which are used to argue: "Since clay minerals contain most of the TEs, we suggest the TE signal directly reflects ocean water oxygen concentrations" (caption to Figure 3). There are two problems here. First, it is not clear to me how clay minerals "containing" the TE signals helps the arguments presented in the paper -- if read at face value, this would suggest that the metals are predominantly detrital, and thus tell you very little about overlying water chemistry. I assume this is not really what the authors mean -- e.g., what they really mean is that the TEs are hosted in spots that appear dominated by clay mineral phases, but there is very little detailed discussion of the laser ablation work in the main text or methods. The authors quote a 30um spot size, and refer to ~100um raster lines, but it is never made clear how each of these were applied, or what is even meant by 'clay minerals' in the in-situ context.

We have included new figures where we substantially address this concern. In these plots, we normalize trace elements (V, U, Mo, plus Ba) to a non-redox sensitive trace element (Sc for V, Mo and Ba and Zr for U) that behaves similarly to trace elements of interest. A plot, then, of this ratio (e.g., V/Sc) against Al shows weak to no correlation. If changes in detrital input alone were responsible for changes in all element concentrations, we would expect the V/Sc, Mo/Sc, U/Zr, and Ba/Sc ratios to vary linearly with Al. They do not, thus we suggest that changes in redox-sensitive trace elements and Ba are due to changes in water column chemistry and productivity, respectively.

In addition, we note that the organic content of these samples are quite low. We suspect that the N is primarily contained within clay minerals, as evidenced by the loose correlation between N and Rb. Since organic matter is low, we interpret that there is little detrital organic matter.

We have used laser ablation analyses to demonstrate that the carbonate fraction does not contribute to trace element whole rock analyses. Thus, by presenting data by correcting for carbonate fraction, we suggest that the detrital influence is small.

Further, neither the V or U enrichments shown are dramatically elevated, and if the Mo content is similarly low it is very likely that much of the metal inventory in these sediments is associated with the detrital load. This would be fully consistent with the authors' contention that the metals are largely hosted with clay minerals, but would strongly mitigate any proxy information for the chemistry of the overlying water column.

We stressed the point that the non-carbonate material contains the TE signal to try and untangle any effects that the detrital input of carbonate could have had. We note that there is correlation between bulk Al and our elements of interest in this study. This is likely because Al and trace elements are all more enriched in the non-carbonate fraction of the samples. This is evident in a plot of V or Al vs residue (i.e., material left after decarbonation).

We now say (lines 133-136):

We suggest that variations in redox sensitive TE could be not due to changes in detrital influence alone. By normalizing TE to non-redox sensitive elements with similar geochemistry (i.e., Sc, Zr) and comparing with a detrital proxy (Al), we show that variations in detrital input, as indicated by Al, cannot fully explain these data (Fig. 5). These plots are corrected for non-carbonate fraction.

I take the authors' point that continued oxidative weathering and trace element delivery may require a sustained O₂ source given the duration of the glaciation. However, I think this argument would be greatly strengthened by a quantitative consideration of how crustal weathering sinks would be expected to scale back during a large-scale glaciation. Just a back-of-the-envelope would do. For example, assuming that crustal weathering ceases entirely, and volcanic reductant input is on the order of, say, ~3 Tmol/y (in O₂ equivalents), at 1PAL the long-term residence time of O₂ in the atmosphere would be on the order of 10 Myr. If atmospheric pO₂ was less, the residence time would go down. So, I don't disagree with the authors' argument here, but I think that it won't be transparently obvious for many readers that the strong source need be required and that a simple calculation will make this more accessible and compelling.

Thank you for this suggestion. We agree that a simple calculation is helpful in illustrating our interpretation that an oxygen source is required during the Snowball glaciation, as lifetime of atmospheric O₂ is shorter than the length of the glacial. We now say around line 140:

Even if oxalic continental weathering were to shut off completely, a modern amount (3.5×10^{19} mol) of O₂ would be completely exhausted in ~ 10 Myr given a volcanic/metamorphic reducing gas flux of 3 Tmol yr⁻¹ [31]. If atmospheric O₂ goes down, and reducing input stays constant, the time needed to deplete atmospheric O₂ time will go down. Given that the residence time of atmospheric oxygen is shorter than the glaciation, a contemporaneous source of oxygen from primary productivity followed by burial of organic matter is required to maintain O₂ levels sufficient for oxidative weathering, as required by TE concentrations.

I'm not sure that Figure 5 is really necessary. It doesn't really contribute any essential or novel information that the text and other figures don't address. Indeed this does not present new information, but it does present a visual summary of our interpretation. As some readers will not be intimately familiar with the N-cycle and its relationship to oxygen, we suggest that this figure does indeed add value to the manuscript.

The final paragraph seems cursory and vague. For example, "Even during periods of global glaciations, organisms procured nutrients and survived" (Lines 192-193) is bordering on tautological given phylogenetic constraints. I think I understand what the authors are going for here, but it needs to be developed a little further and clarified.

We have reworded the final paragraph as such:

The N-cycle and the history of O₂ are intertwined, especially during times of great environmental change or stress. During the Neoproterozoic Marinoan glaciation, all modern N-cycle processes were present as indicated by enriched $\delta^{15}\text{N}$ values in synglacial sediments, though their relative activity was markedly different than modern (Fig. 7). The resilience and flexibility of the N and O₂ cycles, as revealed by TE values, have implications for not only the survival of life on Earth but also for life on other cosmic bodies. The presence of an active biosphere on a nearly entirely frozen planet is indicative of life's propensity to endure.

"Ghuab" is mis-spelled in Line 153.

Typo corrected, thank you.

Reviewer #3 (Remarks to the Author):

Review of Marine primary productivity and oxygen production during Snowball Earth
By Ben Johnson and Colin Goldblatt

This manuscript presents a chemostratigraphic study of Neoproterozoic units in SW Africa deposited during and following the Marinoan glaciation. The problem investigated is whether there is evidence for oxygen production by marine photosynthesizers during this glacial event, which might serve as evidence for less-than-total glaciation of the Earth's oceans (the "hard snowball Earth" model). The data produced are of a high-quality, the manuscript is generally well-written and -illustrated, and the interpretations are mostly reasonable. Overall, I think that this study may be publishable following moderate revision.

Major points:

Conclusions--I am in agreement with the authors' main conclusion that their geochemical results suggest that oxygen production occurred during much of the Marinoan glaciation, and that therefore, a "hard snowball Earth" did not exist for the duration of the glaciation (4-20 Myr). On the other hand, I would like the authors to consider whether a "hard snowball Earth" might have existed for a fraction of the Marinoan glaciation, e.g., during the interval of the SGS, followed by a longer interval of less-complete glaciation. Are their data permissive of this possibility?

While it is possible for a short-duration "hard" Snowball ocean to exist, we suggest that the TE data require a constant supply of O₂ to the atmosphere to facilitate oxidative weathering. If this is shut off, we would expect TE enrichments to disappear over a 2 Myr time period at most. It is therefore most parsimonious to expect continual open water of some extent, though we cannot completely rule out short periods of total ice cover. See line 159:

It is possible that a "hard" Snowball ocean may have existed for a time, but its duration would have to be much shorter than the timescale of oxic weathering (~ 2 Myr) in order to satisfy TE constraints.

Redox interpretations--The authors use trace metal concentrations (or enrichment factors) to evaluate paleoredox conditions in the study units. This is, broadly speaking, OK, and certainly the high EFs of some units suggest strongly reducing conditions. The matter is less clear for the SGS unit, which shows low EFs. These low EFs might indicate oxic watermass conditions, or possibly a reservoir effect (local drawdown of aqueous trace metals) or limited oxic subaerial weathering of trace metals. The uncertainty here would be best resolved through Fe speciation analysis, which would be highly desirable for these units.

Please refer to our Figure 6. We now present the data as whole rock concentrations, corrected for non-carbonate fraction. In addition, elements normalized to Zn (a proxy for organic matter), indicate that the TEs were carried to the sediments with organic matter. Thus, we suggest that the use of EFs is robust. You are correct that local, reservoir effects cannot be ruled out. Certainly Fe speciation would be ideal. We suggest, however, that this new presentation of the data, in conjunction with new treatment of possible detrital influence (Figs. 4-5) present robust evidence for redox-control over TE concentrations.

Ba proxy--It is unclear whether the use of Ba as a paleoproductivity proxy is justified. In the modern open ocean, sediment Ba concentrations reflect productivity because (i) the concentrations of Ba in seawater and sulfate on the

surface of decaying organic particles exceed the solubility product of barite, (ii) there is ample time for barite to accumulate during the ~5000 m sinking of organic particles to the deep-ocean floor, and (iii) barite remains thermodynamically stable under oxic conditions. Potential problems: (1) The study units were presumably not deposited in the open ocean (the text states only "deep water" on lines 76-77, but these are epicratonic units, so water depths were probably no more than a few hundred meters), limiting time for Ba uptake. (2) If the water column was anoxic, then barite would have been thermodynamically unstable, with a tendency to dissolve. (3) In many regions globally, the Neoproterozoic was a time interval of intense hydrothermal activity, and the Ba might have been hydrothermally sourced. The authors could address these uncertainties first by determining what phase the Ba is present in (barite or something else?), and second by investigating whether the elemental chemistry of the study units provides any indication of hydrothermal inputs (e.g., Fe/Mn ratios).

We have updated our discussion of the Ba proxy to account for uncertainties introduced when deposition conditions are anoxic. Specifically, in lines 169-178 we acknowledge that the Ba-proxy may only be a rough indicator of productivity. But, as it does not simply correlate with detrital input (by comparison with Al), we suggest that some variation may be related to changes in productivity, especially in section SGC.

One section (SGS) has a high Fe/Mn ratio (20-120) while the other two sections have Fe/Mn ratios of <10. One might expect a hydrothermal influence to increase the Fe/Mn ratio, as hydrothermal fluids tend to have a high Fe/Mn ratio. An anoxic deep ocean would allow for the transport of this high Fe/Mn ratio fluid. An anoxic (and especially euxinic) deep ocean would, however, promote scavenging of Mo, V, and U close to any hydrothermal vent, and thus would not affect distant sedimentary deposits. We have included these figures in the supplemental material.

Alteration of N isotopes--On lines 102-104 it is stated: "If N were volatilized and lost with progressive metamorphism, concentration and $\delta^{15}\text{N}$ would be negatively correlated, which is not observed (Fig. 2)." In fact, there appears to be a modest negative correlation between [N] and $\delta^{15}\text{N}$ in Figure 2A. The R^2 might be only 0.2-0.3 but given the large number of samples present, this may well be a significant correlation. What is the R^2 value and significance level for the relationship in Figure 2A? Doesn't this in fact suggest that the N-isotope compositions of the study samples may have been affected by deep burial/metamorphic processes?

We suggest that there only appears to be a correlation in Fig. 2 between $\delta^{15}\text{N}$ and [N] when you view all three sections at once. No correlation is seen within an individual section. Apparent correlation is simply the result of different N concentrations in the three sampled sections.

Modern versus Neoproterozoic values--The authors need to exercise some caution in citing modern ocean values for various parameters. These values are not necessarily valid for the Neoproterozoic. For example, the seawater residence times for trace metals cited on lines 123-124.

Thank you, this is a fair point. We have mentioned in these lines that the residence times quoted are for modern waters. During lower O_2 times, residence times would be even shorter. A shorter residence time for any of the TEs used in this study actually strengthens the need for oxic weathering, as lower residence times in the ocean would result in lower TE enrichments in the sections measured.

This issue also arises on Lines 144-145: "Uranium is generally more enriched than

Mo, so the water is not euxinic." This statement depends on seawater having the modern ratio of Mo to U. If, however, the Mo/U ratio of Neoproterozoic seawater were significantly lower than the modern value, then this statement would be incorrect. Again, greater caution is required.

We have added at the end of this sentence: "given a similar dissolved U/Mo ratio to the modern ocean." to clarify this point.

Minor issues:

There are some grammatically incorrect sentences, e.g., lines 127-128, 132-133, 146-147.

Thank you for identifying these mistakes. They have all been corrected. 127-128 now reads:

The present day lifetime of atmospheric oxygen with respect to oxic weathering is ~ 2 Myr [28], and would have been similar or shorter in the Neoproterozoic, when atmospheric oxygen was an order of magnitude lower, due to the rate dependence of oxic weathering on oxygen concentration to a power of 0.5 to 1 [29, 30].

Lines 132-133:

Barium is a productivity proxy in the modern ocean [31]. Barium released during organic matter degradation bonds with sulphate to form barite, which is preserved in oxic sediments.

Lines 146-147:

Barium is highly enriched, which may suggest high productivity, which would mean high oxygen demand. We note again that these are near-shore, and clearly glacially derived. A likely explanation, then, for these geochemical signals is high nutrient input from glacial runoff, supporting high productivity and oxygen use.

"Upsection" is a single word (given incorrectly on lines 92-93). "Synglacial" can be written without a hyphen. So can "nearshore".

We have made these adjustments.

Line 93: "modern ocean bulk $\delta^{15}\text{N}$ ". It would be more accurate to write "modern seawater nitrate $\delta^{15}\text{N}$ ".

Changed, and agreed.

"N-fixing" (line 106 and elsewhere) should be changed to "N fixation".

Changed, and agreed.

Line 120: "between oxic, allowing TE to accumulate". This sentence needs to be refined to "... accumulate in the water column" (as opposed to the sediment). The present wording is ambiguous.

We have changed this sentence to read: "Second, ocean oxidation state must have varied between oxic, allowing TE to accumulate in the water column, and anoxic and in some cases euxinic, when TE were scavenged to the sediments." for clarity.

The authors define three stratigraphic intervals: synglacial siliciclastic (SGS), synglacial carbonate (SGC), and deglacial carbonate (DGC). These intervals are shown and labeled in Figure 4 (although the acronyms are not shown here, but they could be). These intervals probably correspond in some manner to the three groups of sample points shown in Figure 1, but those points are not labeled. The ranges of the SGS, SGC, and DGC units need to be shown clearly in Figure 1.

Thank you, we have added these acronyms to each figure.

Figure 3--are the relationships shown here important? The four graphs just show that trace metals are enriched in the shaly samples (high K₂O) and depleted in the carbonate-rich samples (high CaO+MgO); there is also a small group of samples

located at the origin of all four graphs (and thus probably cherts) that contain near-zero trace metal concentrations. These are completely unsurprising findings, and they could probably be stated in the text without the need for a figure (the figure could be transferred to an SI file, for example).

We suggest that this figure is useful, in that it helps to confirm that the non-carbonate fraction contains the TE budget. As the carbonate detritus was sourced from older units, demonstrating that it is not influencing TE concentrations and our interpretations.

Additionally, we have included a figure (Fig. 5) which compares non-carbonate fraction corrected ratios of elements of interest (U, V, Mo, Ba) normalized to non-redox sensitive elements (Sc, Zr) and Al, which is a detrital input proxy. The lack of correlation in this figure suggests that changes in detrital input alone are not enough to control TE budgets, even though the TE are contained primarily in the clays, which themselves are ultimately of detrital origin. Instead, we suggest that water-column chemistry is a major control on element variation in these units.

Figure 4--Why are TOC and $\delta^{13}C$ profiles not included in this figure? These data are integral to the study's interpretations, so their secular variation should be illustrated in one of the figures.

We did not C-analyses in our interpretation because they appear to be altered, and therefore untrustworthy. The strong, positive correlation between [C]_{org} and C/N seen in Fig. 2 indicates that C loss likely affect changes in the C/N ratio. In addition, estimated metamorphic temperatures for the region (~325°C) are above the point at which C degrades. It is unfortunate that they are altered, as these would be valuable in conjunction with N analyses for interpretation.

Reviewers' comments:

Reviewer #1 (Remarks to the Author):

The authors made a strong and nice effort to answer the request of the first round of review. I still have few comments on this last version. I still have some concern about the detrital origin of the N which could come from non-authigenic clays. I agree with the fact that N seems to be in the clays but what tell the authors that the clays are not detrital?

A recent paper by Paul Hoffman in 2016 (Geobiology) suggest that there was a possible production of organic matter during the glaciation by the cryoconite development. If the authors think that this could be the origin of organic matter buried, a reference to that organic matter would improve and reinforced there statement.

Some minors comments:

Line 84 : Replace Formations by Formation

Line 112 : Replace Ghuab by Ghaub

Line 145 : Could the authors develop why the denitrification should occur only in the sediment and not in a stratified water column ?

Reviewer #2 (Remarks to the Author):

Johnson and Goldblatt have substantively addressed my principal concerns, and I think the paper is an interesting contribution that will stimulate critical discussion. I would be happy to see it published subject to editorial discretion.

Reviewer #3 (Remarks to the Author):

Second review of Marine primary productivity and oxygen production during Snowball Earth (NCOMMS-16-21380A)

1) Trace Metal redox interpretations

The authors use trace metal concentrations (or enrichment speaking, OK, and certainly the high EFs of some units suggest strongly reducing conditions. The matter is less clear for the SGS unit, which shows low EFs. These low EFs might indicate oxic watermass conditions, or possibly a reservoir effect (local drawdown of aqueous trace metals) or limited oxic subaerial weathering of trace metals. The uncertainty here would be best resolved through Fe speciation analysis, which would be highly desirable for these units.

Please refer to our Figure 6. We now present the data as whole rock concentrations, corrected for non-carbonate fraction. In addition, elements normalized to Zn (a proxy for organic matter), indicate that the TEs were carried to the sediments with organic matter. Thus, we suggest that the use of EFs is robust. You are correct that local, reservoir effects cannot be ruled out.

Certainly Fe speciation would be ideal. We suggest, however, that this new presentation of the data, in conjunction with new treatment of possible detrital influence (Figs. 4-5) present robust evidence for redox-control over TE concentrations.

Reviewer: Zn is commonly associated with organic matter but not always, so this is an assumption unless otherwise proven. Further, if the study units were subject to hydrothermal influence (as mentioned in the manuscript), then the Zn may have been hydrothermally sourced rather than related to sediment organic content. So, the Zn-normalization procedure introduced in the revised manuscript is of dubious value.

The authors made no attempt to address my key point: that Fe-speciation data ought to be generated to test their trace-metal-based redox interpretations.

2) Ba proxy

It is unclear whether the use of Ba as a paleoproductivity proxy is justified. In the modern open ocean, sediment Ba concentrations reflect productivity because (i) the concentrations of Ba in seawater and sulfate on the surface of decaying organic particles exceed the solubility product of barite, (ii) there is ample time for barite to accumulate during the ~5000 m sinking of organic particles to the deep-ocean floor, and (iii) barite remains thermodynamically stable under oxic conditions. Potential problems: (1) The study units were presumably not deposited in the open ocean (the text states only “deep water” on lines 76-77, but these are epicratonic units, so water depths were probably no more than a few hundred meters), limiting time for Ba uptake. (2) If the water column

was anoxic, then barite would have been thermodynamically unstable, with a tendency to dissolve. (3) In many regions globally, the Neoproterozoic was a time interval of intense hydrothermal activity, and the Ba might have been hydrothermally sourced. The authors could address these uncertainties first by determining what phase the Ba is present in (barite or something else?), and second by investigating whether the elemental chemistry of the study units provides any indication of hydrothermal inputs (e.g., Fe/Mn ratios).

We have updated our discussion of the Ba proxy to account for uncertainties introduced when deposition conditions are anoxic. Specifically, in lines 169-178 we acknowledge that the Ba-proxy may only be a rough indicator of productivity. But, as it does not simply correlate with detrital input (by comparison with Al), we suggest that some variation may be related to changes in productivity, especially in section SGC.

Reviewer: I remain skeptical that Ba has any validity as a paleoproductivity proxy in these units. One of the other reviewers hammered this point home even more forcefully.

One section (SGS) has a high Fe/Mn ratio (20-120) while the other two sections have Fe/Mn ratios of <10. One might expect a hydrothermal influence to increase the Fe/Mn ratio, as hydrothermal fluids tend to have a high Fe/Mn ratio. An anoxic deep ocean would allow for the transport of this high Fe/Mn ratio fluid. An anoxic (and especially euxinic) deep ocean would, however, promote scavenging of Mo, V, and U close to any hydrothermal vent, and thus would not affect distant sedimentary deposits. We have included these figures in the supplemental material.

Reviewer: I do not agree with the thread of the author's argument here, i.e., that there would not be whole-ocean effects on seawater trace-metal chemistry linked to redox conditions.

3) Alteration of N isotopes

On lines 102-104 it is stated: "If N were volatilized and lost with progressive metamorphism, concentration and $\delta^{15}\text{N}$ would be negatively correlated, which is not observed (Fig. 2)." In fact, there appears to be a modest negative correlation between [N] and $\delta^{15}\text{N}$ in Figure 2A. The R^2 might be only 0.2-0.3 but given the large number of samples present, this may well be a significant correlation. What is the R^2 value and significance level for the relationship in Figure 2A? Doesn't this in fact suggest that the N-isotope compositions of the study samples may have been affected by deep burial/metamorphic processes?

We suggest that there only appears to be a correlation in Fig. 2 between $\delta^{15}\text{N}$ and [N] when you view all three sections at once. No correlation is seen within an individual section. Apparent correlation is simply the result of different N concentrations in the three sampled sections.

Reviewer: The spatial element (i.e., multiple sections) may not be important here. The important point may be that $\delta^{15}\text{N}$ and TN show an overall negative correlation that records changes in N-isotope composition as a function of burial/metamorphic N loss.

4) This issue also arises on Lines 144-145: "Uranium is generally more enriched than Mo, so the water is not euxinic." This statement depends on seawater having the modern ratio of Mo to U. If, however, the Mo/U ratio of Neoproterozoic seawater were significantly lower than the modern value, then this statement would be incorrect. Again, greater caution is required.

We have added at the end of this sentence: "given a similar dissolved U/Mo ratio to the modern ocean." to clarify this point.

Reviewer: To really make the point clear, the authors would need to note that the Mo/U ratio of Neoproterozoic seawater may have been significantly different from the modern value.

Thank you for these reviews. As previously, we have included our responses in black, any previous responses by us in *gray and italics*, and the original reviewer comments in *grey*. The most substantial change is that we have obtained Fe-speciation data and added Dr. Simon Poulton as a co-author. The Fe-speciation data suggests mostly anoxic bottom waters, with some periods of possible oxic bottom waters. Our previous interpretations are corroborated by this new data.

Importantly, we have removed discussion of Ba as a productivity proxy and the normalization of trace element data to Zn as an organic matter proxy, as after further thought we found the data non-compelling. With the addition of Fe-speciation data, however, we have focused our discussion on the redox state of the water, and now demonstrate that the local bottom water was anoxic with oxygenated upper waters and atmosphere.

Overall, the manuscript is shorter and tighter. The message is more focused: there seems to have been anoxic bottom waters with oxygenic photosynthesis, oxidative weathering on the continents, and a N-cycle with fixing, nitrification, and nearly quantitative nitrification.

Reviewers' comments:

Reviewer #1 (Remarks to the Author):

The authors made a strong and nice effort to answer the request of the first round of review. I still have few comments on this last version. I still have some concern about the detrital origin of the N which could come from non-authigenic clays. I agree with the fact that N seems to be in the clays but what tell the authors that the clays are not detrital?

The clays themselves could certainly be detrital. We now show, in the supplemental information, a plot of N/Rb vs Al₂O₃. As with other trace elements, if changes in detrital input of N alone were responsible for variations in N concentration, we would expect a correlation between N/Rb and Al. There is no such correlation, corroborating little detrital influence.

Nitrogen is retained in clays, because after organic matter sinks into the sediment and degrades, N is released as NH₄⁺. This NH₄⁺ then substitutes into clay lattices (Ader et al., 2016 for a review).

A recent paper by Paul Hoffman in 2016 (Geobiology) suggest that there was a possible production of organic matter during the glaciation by the cryoconite development. If the authors think that this could be the origin of organic matter buried, a reference to that organic matter would improve and reinforced there statement.

We think that open water areas best explain the data, but cannot necessarily rule out the cryoconite idea. We have thus cited this in lines 172-175.

Some minors comments:

Line 84 : Replace Formations by Formation
Fixed!

Line 112 : Replace Ghuab by Ghaub
Fixed!

Line 145 : Could the authors develop why the denitrification should occur only in the sediment and not in a stratified water column ?

Denitrification will happen wherever there is NO₃ but low O₂. This will occur most readily at a redox-cline. Such a redox-cline can either be in the water column (stratified water column), the sediment-water interface, or within the sediments. The latter two are more likely under a fully oxygenated water column. We have updated this discussion to (lines 189-195):

“Thus, the choices to explain low delta¹⁵N, but non-zero, values in both SGS and SGC are: fully oxygenated water column with denitrification in the sediments only, or extensive water-column anoxia causing nearly complete, but not total, denitrification. The Fe-speciation data indicate that there was predominantly bottom water anoxia, but TEs and non-atmospheric delta¹⁵N require O₂ production and NO₃⁻. Thus, the low, but non-zero delta¹⁵N values seem to result from either a persistent or periodic chemocline.”

Reviewer #2 (Remarks to the Author):

Johnson and Goldblatt have substantively addressed my principal concerns, and I think the paper is an interesting

contribution that will stimulate critical discussion. I would be happy to see it published subject to editorial discretion.

Thank you for the constructive reviews.

Reviewer #3 (Remarks to the Author):

Please see attachments.

Second review of Marine primary productivity and oxygen production during Snowball Earth (NCOMMS-16-21380A)

1) Trace Metal redox interpretations

The authors use trace metal concentrations (or enrichment speaking, OK, and certainly the high EFs of some units suggest strongly reducing

conditions. The matter is less clear for the SGS unit, which shows low EFs. These low EFs might indicate oxic watermass conditions, or possibly a reservoir effect

(local drawdown of aqueous trace metals) or limited oxic subaerial weathering of trace metals. The uncertainty here would be best resolved through Fe speciation analysis, which would be highly desirable for these units.

Please refer to our Figure 6. We now present the data as whole rock concentrations, corrected for non-carbonate fraction. In addition, elements normalized to Zn (a proxy for organic matter), indicate that the TEs were carried to the sediments with organic matter. Thus, we suggest that the use of EFs is robust. You are correct that local, reservoir effects cannot be ruled out.

Certainly Fe speciation would be ideal. We suggest, however, that this new presentation of the data, in conjunction with new treatment of possible detrital influence (Figs. 4-5) present robust evidence for redox-control over TE concentrations.

Reviewer: Zn is commonly associated with organic matter but not always, so this is an assumption unless otherwise proven. Further, if the study units were subject to hydrothermal influence (as mentioned in the manuscript), then the Zn may have been hydrothermally sourced rather than related to sediment organic content. So, the Zn-normalization procedure introduced in the revised manuscript is of dubious value.

The authors made no attempt to address my key point: that Fe-speciation data ought to be generated to test their trace-metal-based redox interpretations.

We have now generated Fe-speciation data. The majority of samples indicate anoxic, ferruginous bottom water conditions, with several equivocal samples. This is consistent with our interpretations of shallow pockets of oxygenated water and low-O₂ deep water which facilitates TE scavenging and denitrification. Please see lines 120-131.

2) Ba proxy

It is unclear whether the use of Ba as a paleoproductivity proxy is

justified. In the modern open ocean, sediment Ba concentrations reflect

productivity because (i) the concentrations of Ba in seawater and sulfate on the surface of decaying organic particles exceed the solubility product of barite, (ii)

there is ample time for barite to accumulate during the ~5000 m sinking of organic particles to the deep-ocean floor, and (iii)

barite remains thermodynamically

stable under oxic conditions. Potential problems: (1) The study units were

presumably not deposited in the open ocean (the text states only “deep water” on

lines 76-77, but these are epicratonic units, so water depths were probably no more

than a few hundred meters), limiting time for Ba uptake. (2) If the water column

was anoxic, then barite would have been thermodynamically unstable, with a tendency

to dissolve. (3) In many regions globally, the Neoproterozoic was a time interval

of intense hydrothermal activity, and the Ba might have been hydrothermally sourced. The authors could address these uncertainties first by determining what phase the Ba is present in (barite or something else?), and second by investigating whether the elemental chemistry of the study units provides any indication of hydrothermal inputs (e.g., Fe/Mn ratios).

We have updated our discussion of the Ba proxy to account for uncertainties introduced when deposition conditions are anoxic. Specifically, in lines 169-178 we acknowledge that the Ba-proxy may only be a rough indicator of productivity. But, as it does not simply correlate with detrital input (by comparison with Al), we suggest that some variation may be related to changes in productivity, especially in section SGC.

Reviewer: I remain skeptical that Ba has any validity as a paleoproductivity proxy in these units. One of the other reviewers hammered this point home even more forcefully.

One section (SGS) has a high Fe/Mn ratio (20-120) while the other two sections have Fe/Mn ratios of <10. One might expect a hydrothermal influence to increase the Fe/Mn ratio, as hydrothermal fluids tend to have a high Fe/Mn ratio. An anoxic deep ocean would allow for the transport of this high Fe/Mn ratio fluid. An anoxic (and especially euxinic) deep ocean would, however, promote scavenging of Mo, V, and U close to any hydrothermal vent, and thus would not affect distant sedimentary deposits. We have included these figures in the supplemental material.

Reviewer: I do not agree with the thread of the author's argument here, i.e., that there would not be whole-ocean effects on seawater trace-metal chemistry linked to redox conditions.

We have removed discussion of Ba as a productivity proxy after more consideration and in light of strong evidence for anoxic bottom waters from Fe-speciation. The Ba-proxy performs very poorly in low SO₄-waters, which are likely in anoxic settings.

We have expanded the discussion of Fe/Mn in the Supplemental Information. We did not intend to suggest that there would be no effect on trace-metal chemistry due to redox changes. Instead, we suggest that anoxic conditions, which would promote high Fe/Mn ratios, would not greatly influence U, V, and Mo concentrations locally. All three trace elements are much less soluble in anoxic conditions, thus should be scavenged near the hydrothermal vent and be less likely to influence our measured sections.

3) Alteration of N isotopes

On lines 102-104 it is stated: "If N were volatilized and lost with progressive metamorphism, concentration and d15N would be negatively correlated, which is not observed (Fig. 2)." In fact, there appears to be a modest negative correlation between [N] and d15N in Figure 2A. The R² might be only 0.2-0.3 but given the large number of samples present, this may well be a significant correlation. What is the

R² value and significance level for the relationship in Figure 2A? Doesn't this in fact suggest that the N-isotope compositions of the study samples may have been affected by deep burial/metamorphic processes?

We suggest that there only appears to be a correlation in Fig. 2 between d15N and [N] when you view all three sections at once. No correlation is seen within an individual section. Apparent correlation is simply the result of different N concentrations in the three sampled sections.

Reviewer: The spatial element (i.e., multiple sections) may not be important here. The important point may be that d15N and TN show an overall negative correlation that records changes in N-isotope composition as a function of burial/metamorphic N loss.

The observed negative correlation is very weak ($r^2 = 0.137$) between N and $\delta^{15}\text{N}$ for all sections as a whole. If metamorphic loss was the main cause of N concentration variation, we suggest that this correlation would be much stronger.

4) This issue also arises on Lines 144-145: “Uranium is generally more enriched than Mo, so the water is not euxinic.” This statement depends on seawater having the modern ratio of Mo to U. If, however, the Mo/U ratio of Neoproterozoic seawater were significantly lower than the modern value, then this statement would be incorrect. Again, greater caution is required.

We have added at the end of this sentence: “given a similar dissolved U/Mo ratio to the modern ocean.” to clarify this point.

Reviewer: To really make the point clear, the authors would need to note that the Mo/U ratio of Neoproterozoic seawater may have been significantly different from the modern value.

We have added this statement to clarify.

REVIEWERS' COMMENTS:

Reviewer #1 (Remarks to the Author):

Johnson and Goldblatt have substantively addressed my concerns. I am now more convinced and sure this will motivate new studies around these very interesting diamictites.

I thus recommend this final version for publication.

Pierre Sansjofre

Reviewer #3 (Remarks to the Author):

I am satisfied that the authors have addressed each of my major comments adequately. There are a number of omissions in the figures that must be corrected prior to publication (see annotated PDF), but the manuscript is otherwise in good shape now.

Thank you for these final reviews. Original reviewer comments are shown in grey, with our responses shown in black.

REVIEWERS' COMMENTS:

Reviewer #1 (Remarks to the Author):

Johnson and Goldblatt have substantively addressed my concerns. I am now more convinced and sure this will motivate new studies around these very interesting diamictites.

I thus recommend this final version for publication.

Pierre Sansjofre

Thank you for your remarks and comments!

Reviewer #3 (Remarks to the Author):

I am satisfied that the authors have addressed each of my major comments adequately. There are a number of omissions in the figures that must be corrected prior to publication (see annotated PDF), but the manuscript is otherwise in good shape now.

Thank you very much for your review.

We have addressed your final concerns, which include:

- Changing a "*" to "x" in the footnote concerning delta notation
- Inserting spaces before and after all equals-signs
- adding ‰ to Figures 2, 5, and 6